# Regularity as Intrinsic Reward for Free Play

**Cansu Sancaktar**[1] **Justus Piater**[2] **Georg Martius**[1,3]
[1]Max Planck Institute for Intelligent Systems, Germany [2]University of Innsbruck Austria [3]University of Tübingen Germany
cansu.sancaktar@tuebingen.mpg.de

## Abstract

We propose regularity as a novel reward signal for intrinsically-motivated reinforcement learning. Taking inspiration from child development, we postulate that striving for structure and order helps guide exploration towards a subspace of tasks that are not favored by naive uncertainty-based intrinsic rewards. Our generalized formulation of Regularity as Intrinsic Reward (RaIR) allows us to operationalize it within model-based reinforcement learning. In a synthetic environment, we showcase the plethora of structured patterns that can emerge from pursuing this regularity objective. We also demonstrate the strength of our method in a multi-object robotic manipulation environment. We incorporate RaIR into free play and use it to complement the model's epistemic uncertainty as an intrinsic reward. Doing so, we witness the autonomous construction of towers and other regular structures during free play, which leads to a substantial improvement in zero-shot downstream task performance on assembly tasks. Code and videos are available at https://sites.google.com/view/rair-project.

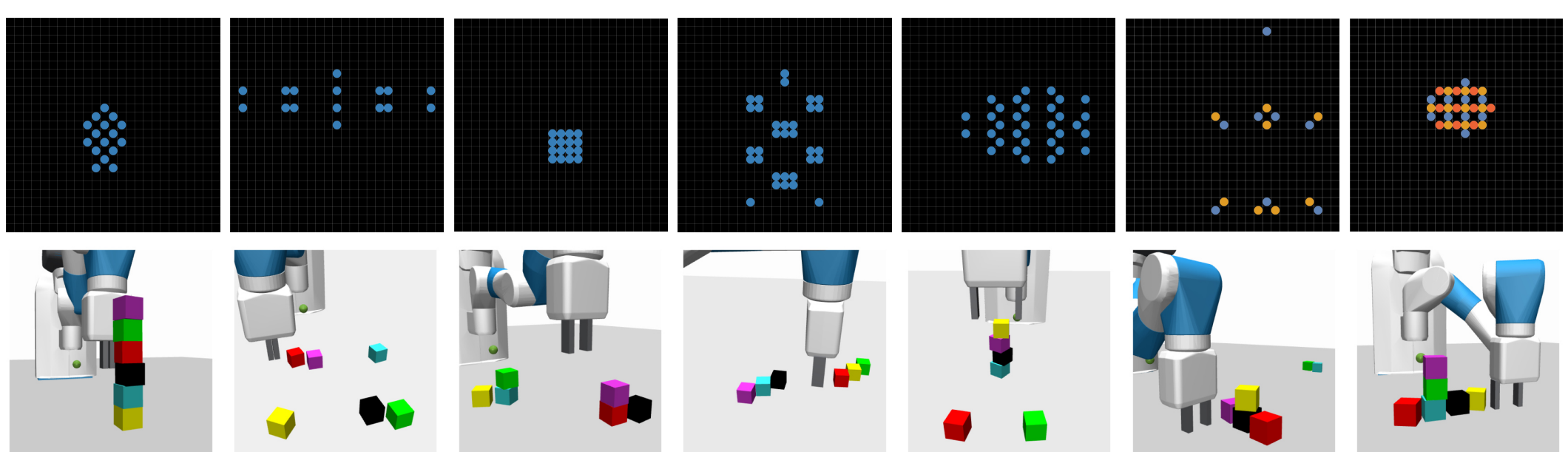

**Figure 1**: **Regularity as intrinsic reward yields ordered and symmetric patterns.** In SHAPEGRIDWORLD (top row) and in CONSTRUCTION (bottom row), we showcase the generated constellations when maximizing our proposed regularity reward RaIR with ground truth (GT) models.

## 1 Introduction

Regularity, and symmetry as a specific form of regularity, are ubiquitous in nature as well as in our manufactured world. The ability to detect regularity helps to identify essential structures, minimizing redundancy and allowing for efficient interaction with the world [1]. Not only do we encounter symmetries in arts, design, and architecture, but our preference also showcases itself in play behavior. Adults and children have both been observed to prefer symmetry in visual perception, where symmetric patterns are more easily detected, memorized, and copied [2, 3]. Several works in developmental psychology show that regular patterns and symmetries are actively sought out during free play in children as well as adults [4–6].

37th Conference on Neural Information Processing Systems (NeurIPS 2023).

Considering this in the context of a child's developmental cycle is intriguing. Studies show that children at the age of 2 exhibit a shift in their exploratory behavior. They progress from engaging in random actions on objects and unstable arrangements to purposefully engaging in functional activities and intentionally constructing stable configurations [7, 8, 5]. Bailey [4] reports that by 5 years of age, children build more structured arrangements out of blocks that exhibit alignment, balance, and examples of symmetries [5].

Despite the dominance of regularity in our perceptual systems and our preference for balance and stability during play, these principles are not yet well investigated within intrinsically-motivated reinforcement learning (RL). One prominent intrinsic reward definition is novelty, i.e. the agent is incentivized to visit areas of the state space with high expected information gain [9–12]. However, one fundamental problem with plain novelty-seeking objectives is that the search space is often unconstrained and too large. As an agent only has limited resources to allocate during play time, injecting appropriate inductive biases is crucial for sample efficiency, good coverage during exploration, and emergence of diverse behaviors. As proposed by Sancaktar et al. [12], using structured world models to inject a relational bias into exploration, yields more object and interaction-related novelty signals. However, which types of information to prioritize are not explicitly encoded in any of these methods. The direction of exploration is often determined by the inherent biases in the practical methods deployed. With imperfect world models that have a limited learning capacity and finite-horizon planning, novelty-seeking methods are observed to prefer "chaotic" dynamics, where small perturbations lead to diverging trajectories, such as throwing, flipping, and poking objects. This in turn means that behaviors focusing on alignment, balance, and stability are overlooked. Not only are these behaviors relevant, as shown in developmental psychology, they also enable expanding and diversifying the types of behavior uncovered during exploration. As the behaviors observed during exploration are highly relevant for being able to solve downstream tasks, a chaos-favoring exploration will make it hard to solve assembly tasks, such as stacking. Indeed, successfully solving assembly tasks with more than 2 objects has been a challenge for intrinsically-motivated reinforcement learning.

We pose the question: how can we define an intrinsic reward signal such that RL agents prefer structured and regular patterns? We propose RaIR: **R**egularity **a**s **I**ntrinsic **R**eward, which aims to achieve highly ordered states. Mathematically, we operationalize this idea using entropy minimization of a suitable state description. Entropy and symmetries have been linked before [13, 14], however, we follow a general notion of regularity, i.e. where patterns reoccur and thus their description exhibits high redundancy / low entropy. In this sense, symmetries are a consequence of being ordered [15]. Regularity also means that the description is compressible, which is an alternative formulation. As argued by Schmidhuber [16], aiming for compression-progress is a formidable curiosity signal, however, it is currently unclear how to efficiently predict and optimize for it.

After studying the design choices in the mathematical formulation of regularity and the relation to symmetry operations, we set out to evaluate our regularity measure in the context of model-based reinforcement learning/planning, as it allows for highly sample-efficient exploration and solving complex tasks zero-shot, as shown in Sancaktar et al. [12]. To get a clear understanding of RaIR, we first investigate the generated structures when directly planning to optimize it using the ground truth system dynamics. A plethora of patterns emerge that are highly *regular*, as illustrated in Fig. 1.

Our ultimate goal is, however, to inject the proposed regularity objective into a free-play phase, where a robot can explore its capabilities in a task-free setting. During this free play, the dynamics model is learned on-the-go. We build on CEE-US [12], a free-play method that uses an ensemble of graph neural networks as a structured world model and the model's epistemic uncertainty as the only intrinsic reward. The epistemic uncertainty is estimated by the ensemble disagreement and acts as an effective novelty-seeking signal. We obtain *structure-seeking free play* by combining the conventional novelty-seeking objective with RaIR.

Our goal is to operationalize regularity, which is a well-established concept in developmental psychology, within intrinsically motivated RL. Furthermore, we showcase that biasing information-search towards regularity with RaIR indeed leads to the construction of diverse regular structures during play and significantly improves zero-shot performance in downstream tasks that also favor regularity, most notably assembly tasks. Besides conceptual work on compression [16, 17], to our knowledge, we are the first to investigate regularity as an intrinsic reward signal, bridging the gap between the diversity of behaviors observed in children's free play and what we can achieve with artificial agents.

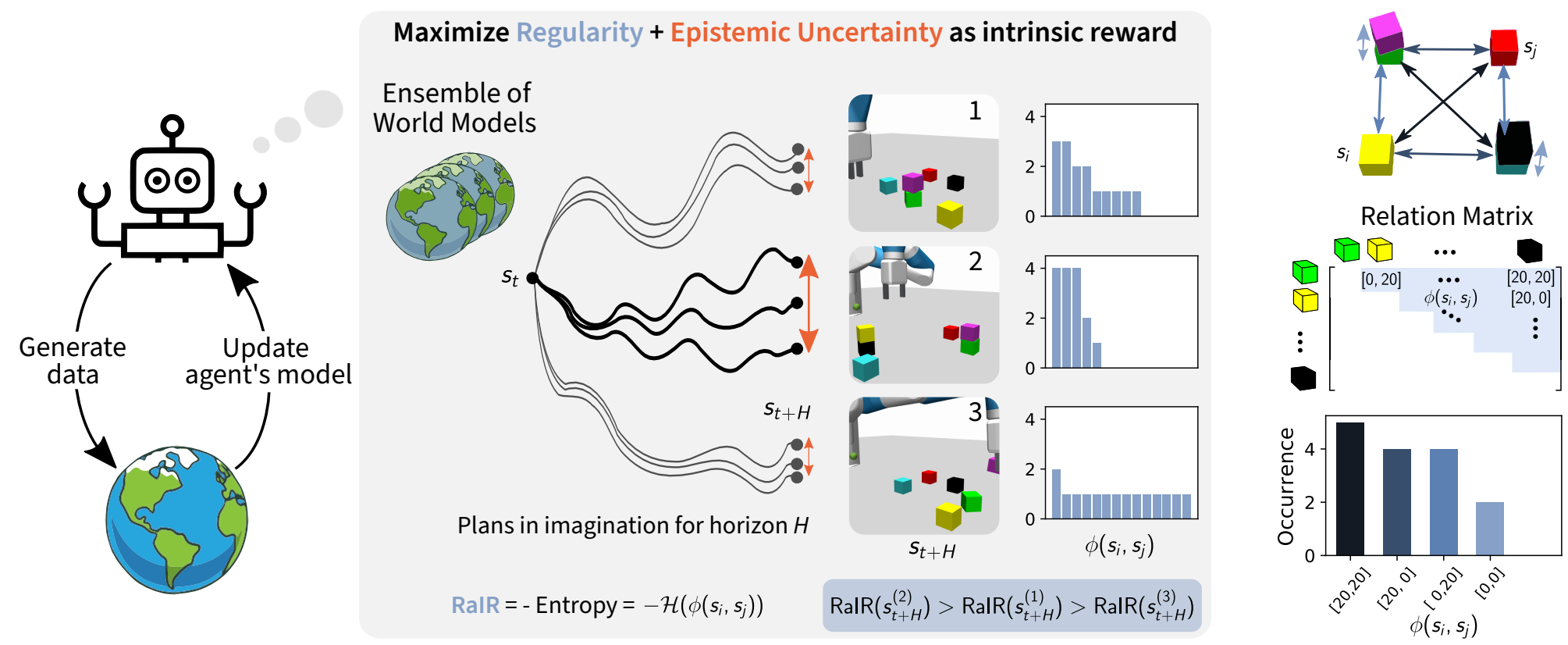

(a) RaIR + CEE-US: Regularity with Epistemic Uncertainty of Structured World Models (b) RaIR computation

**Figure 2**: **Regularity as intrinsic reward during free play.** (a) RaIR + CEE-US uses model-based planning to optimize $H$ timesteps into the future for the combination of RaIR (Eq. 2) and epistemic uncertainty (ensemble disagreement of world models). (b) Here, for RaIR we use the absolute difference vector between objects: $\phi(s_i, s_j) = \{(|\lfloor s_{i,x} - s_{j,x}\rceil|, |\lfloor s_{i,y} - s_{j,y}\rceil|)\}$.

## 2 Method

First, we introduce our intrinsic reward definition for regularity. Then, we present its practical implementation and explain how we combine this regularity objective into model learning within free play.

### 2.1 Preliminaries

In this work, we consider environments that can be described by a fully observable Markov Decision Process (MDP), given by the tuple $(\mathcal{S}, \mathcal{A}, f^a_{ss'}, r^a_{ss'})$, with the state-space $\mathcal{S} \in \mathbb{R}^{n_s}$, the action-space $\mathcal{A} \in \mathbb{R}^{n_a}$, the transition kernel $f : \mathcal{S} \times \mathcal{A} \to \mathcal{S}$, and the reward function $r$. Importantly, we consider the state-space to be factorized into the different entities, e.g. $\mathcal{S} = (\mathcal{S}_{\text{obj}})^N \times \mathcal{S}_{\text{robot}}$ for the state space of a robotic agent and $N$ objects. We use model-based reinforcement learning, where data from interactions with the environment is used to learn a model $\tilde{f}$ of the MDP dynamics [18]. Using this model, we consider finite-horizon ($H$) optimization/planning for undiscounted cumulative reward:

$$\mathbf{a}^\star_t = \arg\max_{\mathbf{a}_t} \sum_{h=0}^{H-1} r(s_{t+h}, a_{t+h}, s_{t+h+1}), \tag{1}$$

where $s_{t+h}$ are imagined states visited by rolling out the actions using $\tilde{f}$, which is assumed to be deterministic. The optimization of Eq. 1 is done with the improved Cross-Entropy Method (iCEM) [19] in a model predictive control (MPC) loop, i.e. re-planning after every step in the environment. Although this is not solving the full reinforcement learning problem (infinite horizon and stochastic environments), it is very powerful in optimizing for tasks on-the-fly and is thus suitable for optimizing changing exploration targets and solving downstream tasks zero-shot.

### 2.2 Regularity as Intrinsic Reward

Quite generally, regularity refers to the situation in which certain patterns reoccur. Thus, we formalize regularity as the **redundancy** in the description of the situation, to measure the degree of sub-structure recurrence. A decisive question is: which description should we use? Naturally, there is certain freedom in this choice, as there are many different coordinate frames. For instance, we could consider the list of absolute object positions or rather a relative representation of the scene.

To formalize, we define a mapping $\Phi : \mathcal{S} \to \{\mathcal{X}\}^+$ from state to a multiset $\{\mathcal{X}\}^+$ of symbols (e.g. coordinates). A multiset is a set where elements can occur multiple times, e.g. $\{a, a, b\}^+$. This multiset can equivalently be described by a tuple $(X, m)$, where $X$ is the set of the unique elements, and $m : X \to \mathbb{Z}^+$ is a function assigning the multiplicity, i.e. the number of occurrences $m(x)$ for

the elements $x \in X$. For the previous example, we get $(\{a, b\}, \{a : 2, b : 1\})$. Given the multiset $(X, m) \in \{\mathcal{X}\}^+$, we define the discrete empirical distribution, also referred to as a histogram, by the relative frequency of occurrence $p(x) = m(x) / \sum_{x' \in X} m(x')$ for $x \in X$.

We define the regularity reward metric using (negative) Shannon entropy [20] of this distribution as:

$$r_{\mathrm{RaIR}}(s) \coloneqq -\mathcal{H}(\Phi(s)) = \sum_{x \in X} p(x) \log p(x) \quad \text{with } (X, m) = \Phi(s), \quad p(x) = \frac{m(x)}{\sum_{x' \in X} m(x')}. \tag{2}$$

We will now discuss concrete cases for the mapping $\Phi$, i.e. how to describe a particular state.

**Direct RaIR.** In the simplest case, we describe the state $s$ directly by the properties of each of the entities. For that, we define the function $\phi : \mathcal{S}_{\text{obj}} \to \{\mathcal{X}\}^+$, that maps each entity to a set of symbols and obtain $\Phi(s) = \uplus_{i=1}^{N} \phi(s_{\text{obj},i})$ as a union of all symbols. The symbols can be, for instance, discretized coordinates, colors, or other properties of the entities.

Let us consider the example where $\phi$ is extracting the object's Cartesian $x$ and $y$ coordinates in a rounded manner as $\phi(s) = \{\lfloor s_x \rceil, \lfloor s_y \rceil\}^+$, as shown in Fig. 3. The most irregular configuration would be when no two objects share the same rounded value in $x$ and $y$. The object configuration becomes more and more regular the more objects share the same $\lfloor x \rceil$ and $\lfloor y \rceil$ coordinates. The most regular configuration is if all objects are in the same place. Note that this choice favors an axis-aligned configuration, and it is not invariant under global rotations.

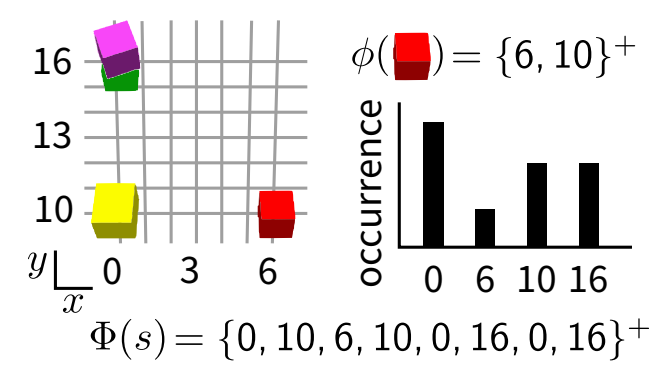

**Figure 3**: Illustration of direct RaIR for $\phi = \{\lfloor x \rceil, \lfloor y \rceil\}$.

**Relational RaIR of order $k$.** Our framework for regularity quantification can easily be extended to a relational perspective, where we don't compute the entropy over aspects of individual entity properties, but instead on their pairwise or higher-order **relations**. This means that for a $k$-order regularity measure, we are interested in tuples of $k$ entities. Thus, the mapping function $\phi$ no longer takes single entities as input, but instead operates on $k$-tuples:

$$\phi : (\mathcal{S}_{\text{obj}})^k \to \{\mathcal{X}\}^+. \tag{3}$$

$\phi$ is a function that describes some relations between the $k$ input entities by a set of symbols.

For $k$-order regularity, the multiset $\Phi$, over which we compute the entropy, is now given by

$$\Phi^{(k)} = \bigcup_{\{i_1,\ldots,i_k\} \in \mathcal{P}} \phi(s_{\text{obj},i_1}, \ldots, s_{\text{obj},i_k}) \quad \text{with } \mathcal{P} = \mathrm{P}(\{1, \ldots, N\}, k) \tag{4}$$

merged from all $k$-permutations of the $N$ entities, denoted as $\mathrm{P}(\{1, \ldots, N\}, k)$. In the case of a permutation invariant $\phi$, Eq. 4 regards only the combinations $\mathrm{C}(\{1, \ldots, N\}, k)$. Note that direct RaIR is equivalent to the relational RaIR of order 1. Given the mapping $\Phi$, the RaIR measure is computed as before with Eq. 2.

Depending on the order $k$ and the function $\phi$, we can select which regularities are going to be favored. Let us consider the example of a pairwise relational RaIR ($k = 2$), where $\phi$ computes the relative positions: $\phi(s_i, s_j) = \{\lfloor s_i - s_j \rceil\}$, and rounding is performed elementwise. Whenever two entities have the same relative position to each other, the redundancy is detected. For $k = 3$ our regularity measure would be able to pick up sub-patterns composed of three objects, such as triangles and so forth.

As we are interested in physical interactions of the robot with objects and objects with objects, we choose RaIR of order $k = 2$ and explore various $\phi$ functions.

### 2.2.1 Properties of RaIR with Pairwise Relations and Practical Implementation

For simplicity, we are considering in the following that $\phi$ maps to a single symbol. Then for pairwise relationships ($k = 2$), RaIR can be implemented using a relation matrix $F \in \mathcal{X}^{N \times N}$. The entries $F_{ij}$ are given by $\phi(s_i, s_j)$ with $s_i, s_j \in S_{\text{obj}}$. After constructing the relation matrix, we need the histogram of occurrences of unique values in this matrix to compute the entropy (Eq. 2). For continuous state spaces, the mapping function needs to implement a discretization step, which we implement by a binning of size $b$. For simplicity of notation, we reuse the rounding notation $\lfloor \cdot \rceil$ for this discretization

**Table 1**: **Properties of RaIR with different $\phi$ regarding symmetry operations**. The first block indicates to which operations RaIR is invariant, ignoring rounding (a.a.: axes aligned). The second block assesses whether a pattern, where the given symmetry operation maps several entities to another, has increased regularity. Rounding and absolute value are elementwise. Distance $d$ is also rounded.

| symmetry operation | invariant? direct $\phi = \lfloor s_i \rceil$ | rel. pos $\lfloor s_i - s_j \rceil$ | \|rel. pos\| $\lvert\lfloor s_i - s_j \rceil\rvert$ | distance $\mathrm{d}(s_i, s_j)$ | favored / increases RaIR? direct $\lfloor s_i \rceil$ | rel. pos $\lfloor s_i - s_j \rceil$ | \|rel. pos\| $\lvert\lfloor s_i - s_j \rceil\rvert$ | distance $\mathrm{d}(s_i, s_j)$ | Sym(∘) = • |
|---|---|---|---|---|---|---|---|---|---|
| translation | ✗ | ✓ | ✓ | ✓ | ✗ | ✓ | ✓ | ✓ | 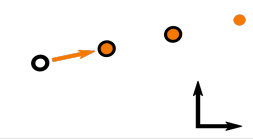 |
| translation – a.a. | ✓ | ✓ | ✓ | ✓ | ✓ | ✓ | ✓ | ✓ | |
| rotation | ✗ | ✓ | ✗ | ✓ | ✗ | ✗ | ✗ | ✓ | 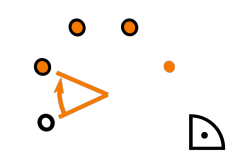 |
| rotation – 90° | ✓ | ✓ | ✓ | ✓ | ✗ | ✗ | ✓ | ✓ | |
| reflection | ✗ | ✓ | ✗ | ✓ | ✗ | ✗ | ✗ | ✓ | 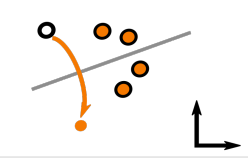 |
| reflection – a.a. | ✓ | ✓ | ✓ | ✓ | ✓ | ✗ | ✓ | ✓ | |
| glide refl. | ✗ | ✓ | ✗ | ✓ | ✗ | ✗$^{(1)}$ | ✗$^{(1)}$ | ✓ | 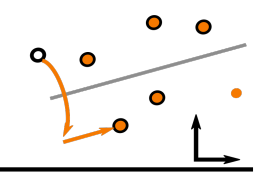 |
| glide refl. – a.a. | ✓ | ✓ | ✓ | ✓ | ✓ | ✗$^{(1)}$ | ✓ | ✓ | |

$^{(1)}$This is for one glide refl. operation. RaIR is increased for 2 glide refl. composition as it collapses onto transl.

step. This bin size $b$ determines the precision of the measured regularity. In practice, we do not apply $\phi$ on the full entity state space, but on a subspace that contains e.g. the $x$-$y$(-$z$) positions.

To understand the properties of our regularity measure for different $\phi$, we present in Table 1 a categorization using the known symmetry operations in 2D and the following $\phi$ (applied to $x$-$y$ positions): direct $\phi(s_i) = \lfloor s_i \rceil$(see previous section), relative position (difference vector) $\phi(s_i, s_j) = \lfloor s_i - s_j \rceil$, absolute value of the relative position[1] $\phi(s_i, s_j) = \lvert\lfloor s_i - s_j \rceil\rvert$, and Euclidean distance $\phi(s_i, s_j) = \lfloor \lVert s_i - s_j \rVert \rceil$. Figure 2b illustrates the RaIR computation using the absolute value of the relative position.

In Table 1, we first consider whether the measure is invariant under symmetry operations. That means if the value of RaIR stays unchanged when the entire configuration is transformed. We find that both Euclidean distance and relative position are invariant to all symmetry operations. The second and possibly more important question is whether a configuration with sub-structures of that symmetry has a higher regularity value than without, i.e. will patterns with these symmetries be favored. We find that Euclidean distance favors all symmetries, followed by absolute value of the relative position. A checkmark in this part of the table means that the more entities can be mapped to each other with the same transformation, the higher RaIR. Although the Euclidean distance seems favorable, we find that it mostly clumps entities and creates fewer alignments. To get a sense of the patterns scoring high in the regularity measure, Fig. 1 showcases situations that emerge when RaIR with absolute value of relative position is optimized (details below).

### 2.3 Regularity in Free Play

Our goal is to explicitly put the bias of regularity into free play via RaIR, as in Fig. 2a. What we want to achieve is not just that the agent creates regularity, but that it gathers valuable experience in creating regularity. This ideally leads to directing exploration towards patterns/arrangements that are novel.

We propose to use RaIR to augment plain novelty-seeking intrinsic rewards, in this work specifically ensemble disagreement [10]. We choose ensemble disagreement because 1) we need a reward definition that allows us to predict future novelty, such that we can use it inside model-based planning (this constraint makes methods relying on retrospective novelty such as Intrinsic Curiosity Module (ICM) [9] ineligible), and 2) we want to use the models learned during free play for zero-shot downstream task generalization via planning in a follow-up extrinsic phase. It has been shown in previous works that guiding exploration by the model's own epistemic uncertainty, approximated

[1]The rounding and the absolute value functions are applied coordinate-wise.

via ensemble disagreement, leads to learning more robust world models compared to e.g. Random Network Distillation (RND) [21] (Suppl. J.1), resulting in improved zero-shot downstream task performance [12]. That is why we choose ensemble disagreement to compute expected future novelty.

We train an ensemble of world models $\{(\tilde{f}_{\theta m})_{m=1}^{M}\}$, where $M$ denotes the ensemble size. The model's epistemic uncertainty is approximated by the disagreement of the ensemble members' predictions. The disagreement reward is given by the trace of the covariance matrix [12]:

$$r_{\text{Dis}} = \text{tr}\big(\text{Cov}(\{\hat{s}_{t+1}^{m} = \tilde{f}_{\theta m}(s_t, a_t) \mid m = 1, \dots, M\})\big). \tag{5}$$

We incorporate our regularity objective into free play by using a linear combination of RaIR and ensemble disagreement. Overall, we have the intrinsic reward:

$$r_{\text{intrinsic}} = r_{\text{RaIR}} + \lambda \cdot r_{\text{Dis}}, \tag{6}$$

where $\lambda$ controls the trade-off between regularity and pure epistemic uncertainty.

**Model-based Planning with Structured World Models** To optimize the reward function on-the-fly, we use model-based planning using zero-order trajectory optimization, as introduced in Sec. 2.1. Concretely, we use CEE-US [12], which combines structured world models and epistemic uncertainty (Eq. 5) as intrinsic reward. The structured world models are ensembles of message-passing Graph Neural Networks (GNNs) [22], where each object corresponds to a node in the graph. The node attributes $\{s_{t,i} \in \mathcal{S}_{\text{obj}} \mid i = 1, \dots, N\}$ are the object features such as position, orientation, and velocity at time step $t$. The state representation of the actuated agent $s_{\text{robot}} \in \mathcal{S}_{\text{robot}}$ similarly contains position and velocity information about the robot. We treat the robot as a global node in the graph [12]. We refer to the combination of RaIR with ensemble disagreement, medium-horizon planning (20-30 time steps), and structured world models as RaIR + CEE-US.

## 3 Experiments

We evaluate RaIR in the two environments shown in Fig. 1.

**ShapeGridWorld** is a grid environment, where each circle represents an entity/agent that is controlled separately in $x$-$y$ directions. Entities are controlled one at a time. Starting from timestep $t = 0$, the entity with $i = 1$ is actuated for $T$ timesteps, where $T$ is the entity persistency. Then, at $t = T$, actuation switches over to entity $i = 2$ and we keep iterating over the entities in this fashion. Each circle is treated as an entity for the regularity computation with a 2D-entity state space $S_{\text{obj}}$ with $x$-$y$ positions.

**Fetch Pick & Place Construction** is an extension of the Fetch Pick & Place environment [23] to more cubes [24] and a large table [12]. An end-effector-controlled robot arm is used to manipulate blocks. The robot state $S_{\text{robot}} \in \mathbb{R}^{10}$ contains the end-effector position and velocity, and the gripper's state (open/close) and velocity. Each object's state $S_{\text{obj}} \in \mathbb{R}^{12}$ is given by its pose and velocities. For free play, we use 6 objects and consider several downstream tasks with varying object numbers.

### 3.1 Emerging Patterns in SHAPEGRIDWORLD and CONSTRUCTION with RaIR

To get a sense of what kinds of patterns emerge following our regularity objective with RaIR, we do planning using ground truth (GT) models, i.e. with access to the true simulator itself for planning. We perform these experiments to showcase that we can indeed get *regular* constellations with our proposed formulation. Since we can perform multi-horizon planning without any accumulating model errors using ground truth models, we can better investigate the global/local optima of our regularity reward. Note that as we are using a zero-order trajectory optimizer with a limited sample budget and finite-horizon planning, we don't necessarily converge to the global optima. We use $\phi(s_i, s_j) = \{(|\lfloor s_{i,x} - s_{j,x} \rceil|, |\lfloor s_{i,y} - s_{j,y} \rceil|)\}$ for RaIR in both environments. The emerging patterns are shown in Fig. 1.

In the 2D SHAPEGRIDWORLD environment, we indeed observe that regular patterns with translational, reflectional (axis-aligned), glide-reflectional (axis-aligned), and rotational symmetries emerge. Regularity is not restricted to spatial relations and can be applied to any set of symbols. To showcase this, we apply RaIR to colored SHAPEGRIDWORLD, where color is part of $S_{\text{obj}}$. Generated patterns are not just regular in $x$-$y$ but also in color, as shown in Fig. S7 and the 2 top right examples in Fig. 1.

For CONSTRUCTION, we also observe complex constellations with regularities, even stacks of all 6 objects. Since we are computing RaIR on the $x$-$y$ positions, a stack of 6 is the global optimum. The optimization of RaIR for this case is shown in Fig. 4. Note that stacking itself is a very challenging task, and was so far only reliably achievable with reward shaping or tailored learning curricula [24]. The fact that these constellations appear naturally from our regularity objective, achievable with a planning horizon of 30 timesteps, is by itself remarkable.

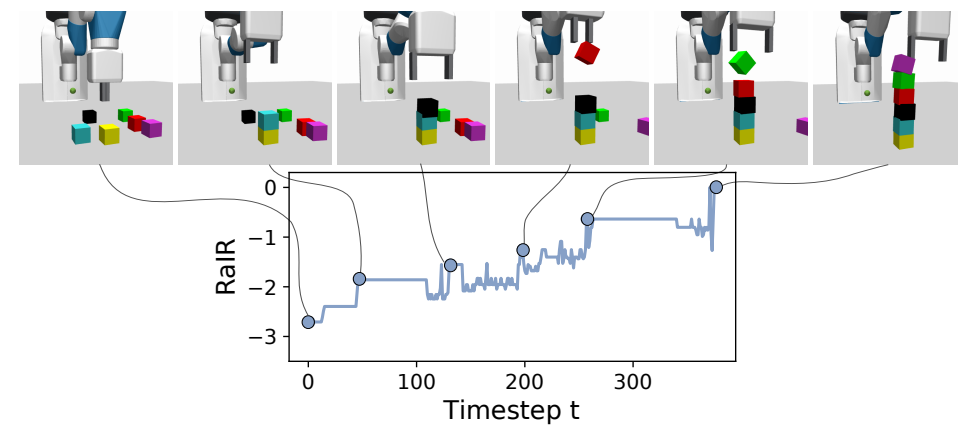

**Figure 4**: **RaIR throughout a rollout** starting from a random initial configuration when optimizing only for regularity with the GT model.

Additional example patterns generated in CONSTRUCTION with RaIR on the $x$-$y$-$z$ positions can be found in the Suppl. A. In that case, a horizontal line on the ground and a vertical line into air, i.e. a stack, are numerically equivalent with respect to RaIR. Choosing to operate on the $x$-$y$-subspace is injecting the direction of gravity and provides a bias towards vertical alignments. We also apply RaIR to a custom CONSTRUCTION environment with different shapes and masses (cubes, columns, balls and flat blocks) and once again observe regular arrangements, as in Fig. S8. More details in Suppl. F.

### 3.2 Free Play with RaIR in CONSTRUCTION

We perform free play in CONSTRUCTION, i.e. only optimize for intrinsic rewards, where we learn models on-the-go. During free play, we start with randomly initialized models and an empty replay buffer. Each iteration of free play consists of data collection with environment interactions (via online planning), and then model training on the collected data so far (offline).

In each iteration of free play, we collect 2000 samples (20 rollouts with 100 timesteps each) and add them to the replay buffer. During the online planning part for data collection, we only perform inference with the models and no training is performed. Afterwards, we train the model for a fixed number of epochs on the replay buffer. We then continue with data collection in the next free play iteration. More details can be found in Suppl. H. For this intrinsic phase, we combine our regularity objective with ensemble disagreement as per Eq. 6. The goal is to bias exploration and the search for information gain towards regular structures, corresponding to the optima that emerge with ground truth models, as shown in Fig. 1. We also show results for the baselines RND [21] and Disagreement (Dis) [10], using the same model-based planning backbone as CEE-US (see Suppl. J). The Dis baseline also uses ensemble disagreement as intrinsic reward, however unlike CEE-US, only plans for one step into the future.

In Figure 5, we analyze the quality of data generated during free play, in terms of observed interactions, for RaIR + CEE-US with the augmentation weight $\lambda = 0.1$, a pure RaIR run with no information-gain component in the intrinsic reward ($\lambda = 0$), CEE-US, as well as RND and Disagreement. For pure RaIR, we observe a decrease in the generated interactions. This has two reasons: 1) RaIR only aims to generate structure and the exploration problem is not solved, 2) once the controller finds a plan that leads to an optimum, even if it is local, there is no incentive to destroy

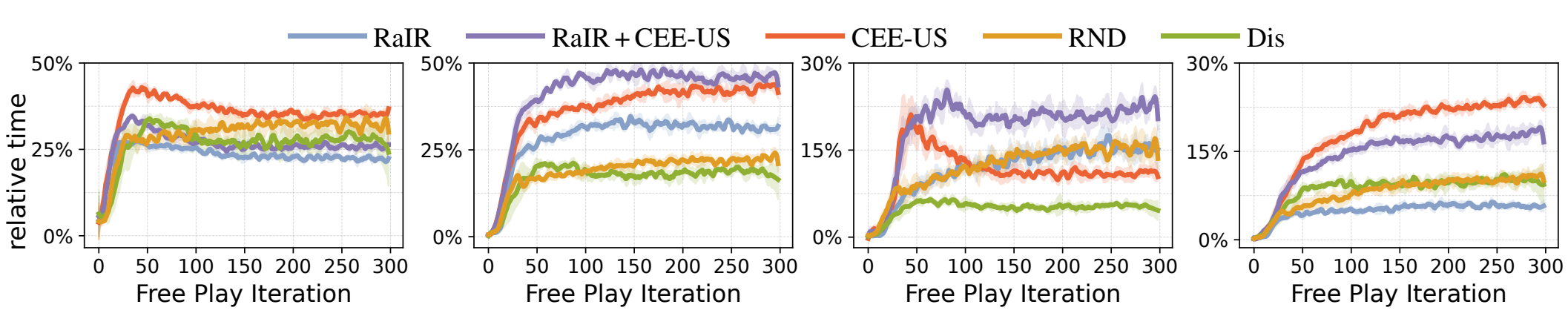

(a) 1 object moves (b) 2 or more objects move (c) object(s) in air (d) object(s) flipped

**Figure 5**: **Comparison of interactions during free play in CONSTRUCTION when combining ensemble disagreement with RaIR (with $\lambda = 0.1$) compared to CEE-US, pure RaIR, RND and Dis.** These metrics count the relative amount of timesteps the agent performs certain types of interactions in the 2K transitions collected at each free play iteration. (a) *1 object moves* checks the amount of time the agent spends moving only one object. (b) *2 or more objects move* checks if at least 2 objects are moving at the same time. (c) *Object(s) in air* means one or more objects are in air (including being held in air by the agent or being on top of another block). (d) *Object(s) flipped* checks for angular velocities above a threshold for one or more objects, i.e. if they are rolled/flipped. We used 5 independent seeds.

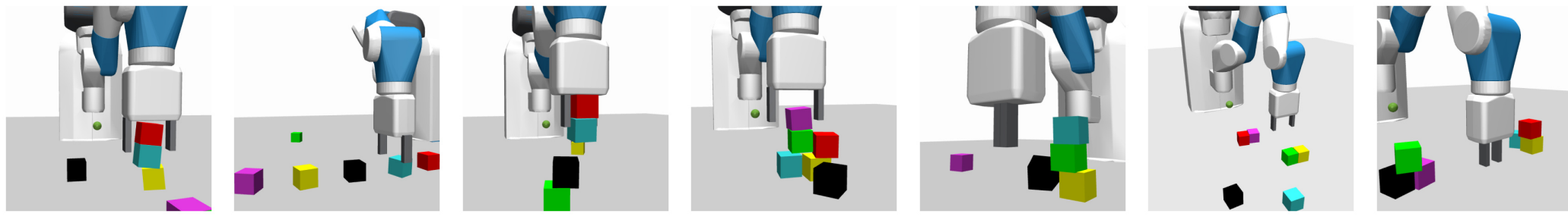

Iteration 251 Iteration 255 Iteration 259 Iteration 275 Iteration 284 Iteration 286 Iteration 296

**Figure 6**: **Snapshots from free play with RaIR + CEE-US.** We showcase snapshots of highest RaIR values, equivalent to lowest entropy, from exemplary rollouts at different iterations of free play. Following the regularity objective, stacks and alignments are generated.

it, unless a plan that results in better regularity can be found within the planning horizon. There is no discrimination between "boring" and "interesting" patterns with respect to the model's current capabilities. This in turn means that the robot creates e.g. a (spaced) line, which is a local optimum for RaIR, and then spends the rest of the episode, not touching any objects to keep the created alignment intact. With the injection of some disagreement in RaIR + CEE-US, we observe improved interaction metrics throughout free play in terms of 2 or more object interactions and objects being in the air (either being lifted by the robot or being stacked on top of another block). In practice, since the ensemble of models tends to hallucinate due to imperfect predictions, even for pure RaIR we observe dynamic pattern generations, as reflected in the interaction metrics (more details in Suppl. C). For the plain disagreement case with CEE-US, more flipping behavior, and less air time are observed during free play, since the agent favors chaos. We also observe that the baselines RND and Disagreement (planning horizon 1) produce less interaction-rich data during free play. Especially for Disagreement, this further shows that planning for future novelty is an essential component for free play.

Another reason why disagreement in RaIR + CEE-US is helpful is due to the step-wise landscape of RaIR as shown in Fig. 4. Here, combining RaIR with ensemble disagreement effectively helps smoothen this reward function, making it easier to find plans with improvements in regularity with imperfect world models.

In Fig. 7, we report the highest achieved RaIR value in the collected rollouts throughout free play. We observe that pure RaIR and RaIR + CEE-US indeed find more regular structures during play compared to the baselines. Some snapshots of regular structures observed during a RaIR + CEE-US free play run are illustrated in Fig. 6. Results for $\phi(s_i, s_j) = \lfloor s_i - s_j \rceil$ can be found in Suppl. B.

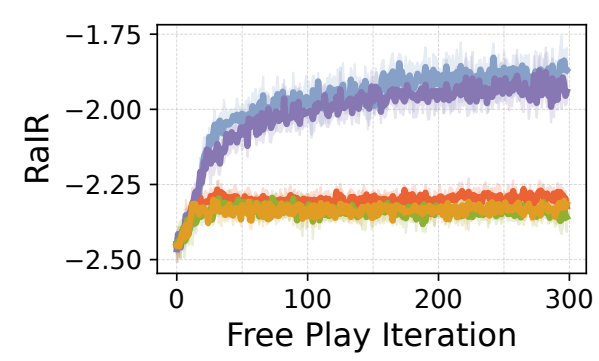

**Figure 7**: **Highest RaIR value throughout free play** for pure RaIR, RaIR + CEE-US, CEE-US, RND and Dis.

### 3.3 Zero-shot Generalization to Assembly Downstream Tasks with RaIR in CONSTRUCTION

After the fully-intrinsic free-play phase, we evaluate zero-shot generalization performance on downstream tasks, where we perform model-based planning with the learned world models. Note that now instead of optimizing for intrinsic rewards, we are optimizing for extrinsic reward functions $r_{\text{task}}$ given by the environment (Suppl. I.4.1). In Fig. 8, we present the evolution of success rates of models checkpointed throughout free play on the following assembly tasks: singletower with 3 objects, 2 multitowers with 2 objects each, pyramid with 5 and 6 objects.

The combination RaIR + CEE-US yields significant improvements in the success rates of assembly tasks, as shown in Fig. 8 and Table 2. RaIR alone, outperforms CEE-US in assembly tasks. As we are biasing exploration towards regularity, we see a decrease in more chaotic interactions during play time, which is correlated with a decrease in performance for the more chaotic throwing and flipping tasks. For the generic Pick & Place task, we observe comparable performance between RaIR + CEE-US and CEE-US. The decrease in performance for RaIR in non-assembly tasks shows the importance of an information-gain objective. The baselines RND and Disagreement exhibit very poor performance in the assembly tasks. In the other tasks, RND and CEE-US are comparable (bold numbers show statistical indistinguishability from best with $p > 0.05$) This showcases that guiding free play by the model's epistemic uncertainty as in the case with ensemble disagreement, helps learn robust and capable world models. The decrease in the zero-shot performance for Disagreement further proves the importance of planning for future novelty during free play. We also run free play in the custom CONSTRUCTION environment for RaIR + CEE-US and CEE-US and observe improved zero-shot downstream performance in assembly tasks with RaIR + CEE-US (see Suppl. F.2).

**Table 2**: **Zero-shot downstream task performance of RaIR + CEE-US, RaIR, CEE-US, RND and Dis** for assembly tasks as well as the generic pick & place task and the more chaos-oriented throwing and flipping. Results are shown for five independent seeds. In the bottom row, we report the success rates achieved via planning with ground truth models. This is to provide a baseline for how hard the task is to solve with finite-horizon planning and potentially suboptimally designed task rewards.

| | Singletower 3 | Multitower 2+2 | Pyramid 5 | Pyramid 6 | Pick&Place 6 | Throw 4 | Flip 4 |
|---|---|---|---|---|---|---|---|
| RaIR + CEE-US | $\mathbf{0.75 \pm 0.07}$ | $\mathbf{0.77 \pm 0.06}$ | $\mathbf{0.49 \pm 0.06}$ | $\mathbf{0.18 \pm 0.04}$ | $\mathbf{0.90 \pm 0.02}$ | $0.32 \pm 0.02$ | $0.63 \pm 0.08$ |
| RaIR | $0.64 \pm 0.03$ | $0.62 \pm 0.03$ | $0.25 \pm 0.05$ | $0.10 \pm 0.02$ | $0.74 \pm 0.05$ | $0.21 \pm 0.01$ | $0.65 \pm 0.1$ |
| CEE-US | $0.40 \pm 0.12$ | $0.52 \pm 0.05$ | $0.14 \pm 0.09$ | $0.02 \pm 0.01$ | $\mathbf{0.90 \pm 0.02}$ | $\mathbf{0.49 \pm 0.05}$ | $\mathbf{0.73 \pm 0.1}$ |
| RND | $0.07 \pm 0.07$ | $0.14 \pm 0.12$ | $0.02 \pm 0.02$ | $0.0 \pm 0.0$ | $\mathbf{0.91 \pm 0.01}$ | $\mathbf{0.42 \pm 0.07}$ | $\mathbf{0.82 \pm 0.1}$ |
| Dis | $0.0 \pm 0.0$ | $0.01 \pm 0.01$ | $0.0 \pm 0.0$ | $0.0 \pm 0.0$ | $\mathbf{0.89 \pm 0.01}$ | $0.30 \pm 0.04$ | $\mathbf{0.69 \pm 0.09}$ |
| GT | 0.99 | 0.97 | 0.82 | 0.81 | 0.99 | 0.97 | 1.0 |

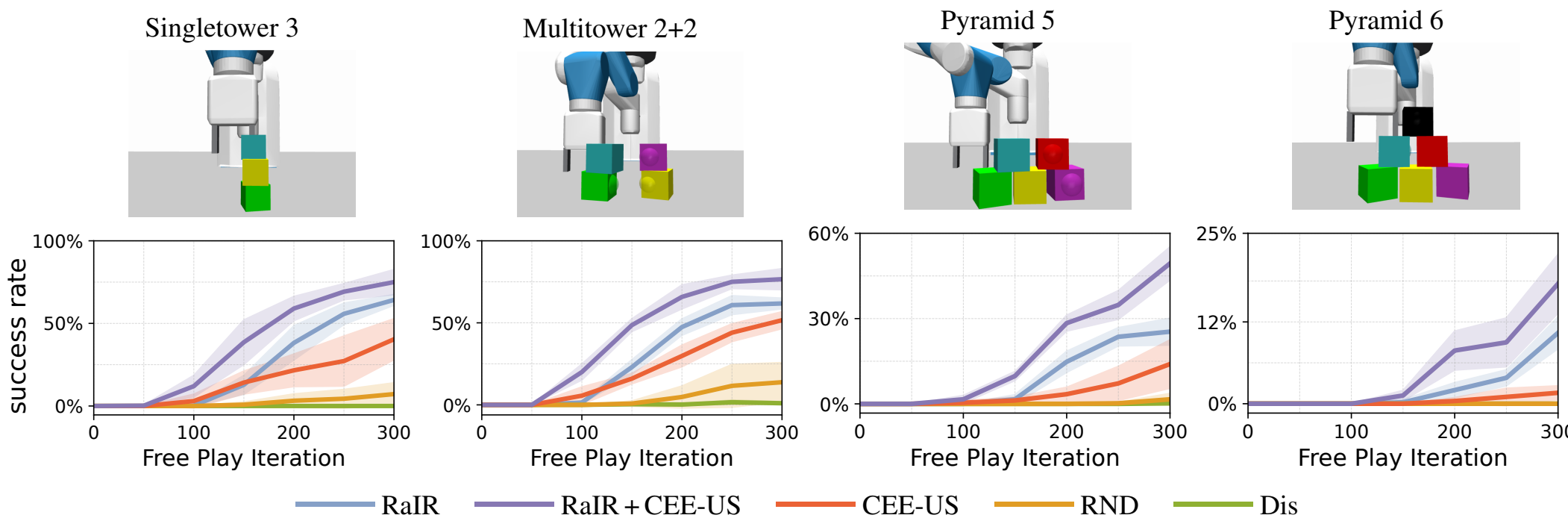

**Figure 8**: **Success rates for zero-shot downstream task generalization for assembly tasks** in CONSTRUCTION for model checkpoints over the course of free play. We compare RaIR + CEE-US $(\lambda = 0.1)$ and RaIR with CEE-US, RND and Dis. We used five independent seeds.

### 3.4 Re-creating existing structures with RaIR

We test whether we can re-create existing arrangements in the environment with RaIR. If there are regularities / sub-structures already present in the environment, then completing or re-creating these patterns naturally becomes an optimum for RaIR, as repeating this pattern introduces redundancy. We initialize pyramids, single- and multitowers out of the robot's manipulability range in CONSTRUCTION. We then plan using iCEM to maximize RaIR with GT models. Doing so, the agent manages to re-create the existing structures in the environment with the blocks it has within reach. In Fig. 9, this is showcased for a pyramid with 3 objects, where in 15 rollouts a pyramid is recreated in 73% of the cases. Without the need to define any explicit reward functions, we can simply use our regularity objective to mimic existing ordered constellations. More details can be found in Suppl. D.

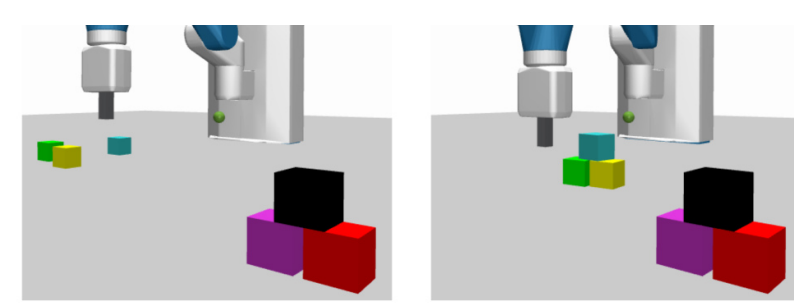

(a) $t = 0$ (b) $t = 200$

**Figure 9**: A pyramid initialized out of the robot's reach is re-created by optimizing for RaIR.

## 4 RaIR in Locomotion Environments

The only requirement to incorporate RaIR in a given environment is an entity-based view of the world. We can easily apply this principle to locomotion environments, where we treat each joint as an entity. In Figure 10, we showcase the generated poses in the DeepMind Control Suite environments Quadruped and Walker, when we optimize for regularity using ground truth models. Here, regularity is computed over the $x$-$y$-$z$ positions of the knees and toes of the quadruped. For walker, we take the positions of the feet, legs and the torso. These poses also heavily overlap with the goal poses proposed in the RoboYoga benchmark [25] (see Fig. S11), which further supports our hypothesis that regularities and our preference for them are ubiquitious. We also apply free play with learned models in the Quadruped environment. More details and results can be found in Suppl. G.

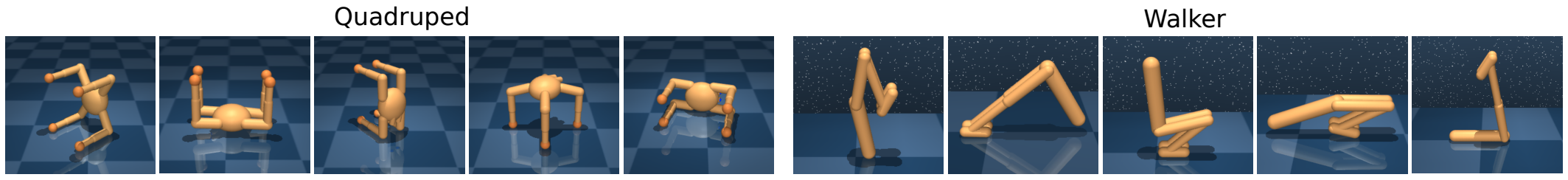

**Figure 10**: **RaIR in Quadruped and Walker environments with GT models.** We show generated poses when maximizing for regularity over the positions of different joints (e.g. knees, toes).

## 5 Related Work

**Intrinsic motivation in RL** uses minimizing novelty/surprise to dissolve cognitive disequilibria as a prominent intrinsic reward signal definition [9, 26–31]. As featured in this work, using the disagreement of an ensemble of world models as an estimate of expected information gain is a widely-used metric as it allows planning into the future [10–12]. Other prominent intrinsic rewards deployed in RL include learning progress [26, 30, 32], empowerment [33, 34] and maximizing for state space coverage with count-based methods [35, 36] and RND [21]. Another sub-category would be goal-conditioned unsupervised exploration methods combined with e.g. ensemble disagreement [25, 37] or asymmetric self-play [38]. In competence-based intrinsic motivation, unsupervised skill discovery methods aim to learn policies conditioned on a latent skill variable [39, 40]. Berseth et al. [41] propose surprise minimization as intrinsic reward to seek familiar states in unstable environments with an active source of entropy. Note that this differs from our work, as our notion of regularity is decoupled from surprise: RaIR aims to get to a state that is regular in itself.

**Compression** and more specifically compression progress have been postulated as driving forces in human curiosity by Schmidhuber [16]. However, the focus has been on *temporal* compression, where it is argued that short and simple explanations of the past make long-horizon planning easier. In our work, we don't focus on compression in the temporal dimension, i.e. sequences of states. Instead, we perform compression as entropy minimization (in the relational case, equivalent to lossy compression) at a given timestep $t$, where we are interested in the relational redundancies in the current scene. More details on connections to compression can be found in Appendix. L.

**Assembly Tasks in RL** with 3+ objects pose an open challenge, where most methods achieve stacking via tailored learning curricula with more than 20 million environment steps [24, 42], expert demonstrations [43], also together with high-level actions [44]. Hu et al. [37] manage to solve 3-object stacking in an unsupervised setting with goal-conditioned RL (GCRL), using a very similar robotic setup to ours, but only with 30% success rate. More discussion on GCRL in Suppl. K.

## 6 Discussion

Although the search for regularity and symmetry has been studied extensively in developmental psychology, these concepts haven't been featured within reinforcement learning yet. In this work, we propose a mathematical formulation of regularity as an intrinsic reward signal and operationalize it within model-based RL. We show that with our formulation of regularity, we indeed manage to create regular and symmetric patterns in a 2D grid environment as well as in a challenging compositional object manipulation environment. We also provide insights into the different components of RaIR and deepen the understanding of the types of regularities emerging from using different mappings $\phi$. In the second part of the work, we incorporate RaIR within free play. Here, our goal is to bias information-search during exploration towards regularity. We provide a proof-of-concept that augmenting epistemic uncertainty-based intrinsic rewards with RaIR helps exploration for symmetric and ordered arrangements. Finally, we also show that our regularity objective can simply be used to imitate existing regularities in the environment.

**Limitations and future work:** As we use finite-horizon planning, we don't necessarily converge to global optima. This can both be seen as a limitation and a feature, as it naturally allows us to obtain different levels of regularity in the generated patterns. Currently, we are restricted to fully-observable MDPs. We embrace object-centric representations as a suitable inductive bias in RL, where the observations per object (consisting of poses and velocities) are naturally disentangled (more discussion in Appendix. M). We also assume that this state space is interpretable such that we take, for instance, only the positions and color. The representational space, in which the RaIR measure is computed, is specified by the designer. Exciting future work would be to learn a representation under which the human relevant structures in the real-world (e.g. towers, bridges) are indeed regular.

## Acknowledgments and Disclosure of Funding

The authors thank Sebastian Blaes, Anselm Paulus, Marco Bagatella, Núria Armengol Urpí and Maximilian Seitzer for helpful discussions and for their help reviewing the manuscript. The authors thank the International Max Planck Research School for Intelligent Systems (IMPRS-IS) for supporting Cansu Sancaktar. Georg Martius is a member of the Machine Learning Cluster of Excellence, EXC number 2064/1 – Project number 390727645. We acknowledge the financial support from the German Federal Ministry of Education and Research (BMBF) through the Tübingen AI Center (FKZ: 01IS18039B). This work was supported by the Volkswagen Stiftung (No 98 571).

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

# Supplementary Material for
# Regularity as Intrinsic Reward for Free Play

Code and videos are available at https://sites.google.com/view/rair-project.

## A Experiment Results with Ground Truth Models

### A.1 Experiment Results for RaIR in CONSTRUCTION with $x$-$y$-$z$

As discussed in Sec. 3.1, in our experiments we compute RaIR on the $x$-$y$ subspace of the object positions in CONSTRUCTION to inject a bias towards vertical alignments. Examples of patterns generated when optimizing for RaIR using

$$\phi(s_i, s_j) = \{(|\lfloor s_{i,x} - s_{j,x}\rceil|, |\lfloor s_{i,y} - s_{j,y}\rceil|, |\lfloor s_{i,z} - s_{j,z}\rceil|)\}$$

are showcased in Fig. S1. When we also include the $z$-positions of the objects in the RaIR computation, patterns and constellations on the ground are preferred. In this case, there is no difference between a horizontal line on the ground vs. a vertical line, i.e. a stack. Since creating a stack, however, is a more sparse solution, in practice the zero-order trajectory optimizer converges already to regular structures on the ground and vertical constellations don't emerge. Starting from a randomly initialized scene with all objects on the ground, the regularity metric for $x$-$y$-$z$ only starts increasing when multiple objects are in the stack, which would require a very long planning horizon to find this solution.

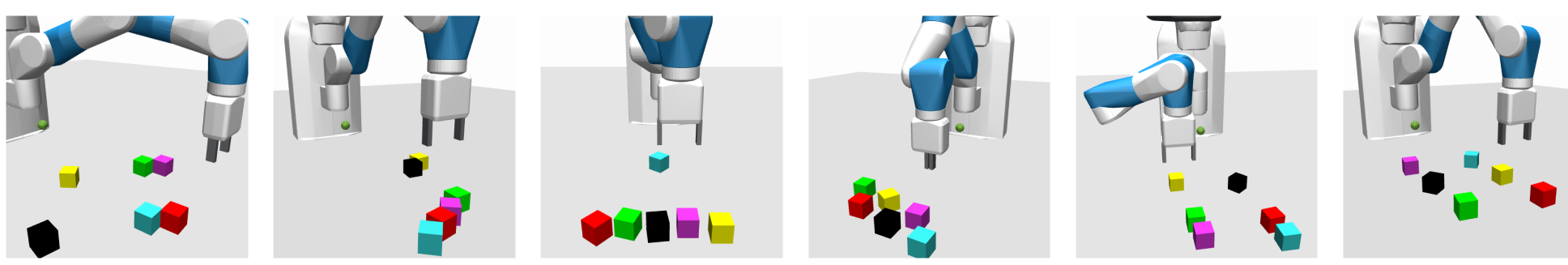

**Figure S1**: **Emerging patterns with RaIR on the $x$-$y$-$z$ subspace with GT models**, where we use absolute relational $\phi$.

## B Experiment Results for Relational Case without Absolute Value

We present the interaction metrics observed during free play with RaIR + CEE-US in the case of relational $\phi$ with $\phi(s_i, s_j) = \{\lfloor s_i - s_j\rceil\} = \{(\lfloor s_{i,x} - s_{j,x}\rceil, \lfloor s_{i,y} - s_{j,y}\rceil)\}$. We find the interaction metrics to be comparable to the absolute relational case presented in the main paper with $\phi(s_i, s_j) = \{(|\lfloor s_{i,x} - s_{j,x}\rceil|, |\lfloor s_{i,y} - s_{j,y}\rceil|)\}$. In Fig. S2, we also include the results for RaIR + CEE-US with $\lambda = 1$. In the case of the increased weighting on the ensemble disagreement term, the free-play behavior indeed collapses back onto CEE-US. This means we have more flipping behavior and less air time. In the case of RaIR + CEE-US with smaller $\lambda = 0.1$, we seek regular states, which include vertical alignments, such that the air time doesn't go down. Note that in this case, there is still an incentive to "destroy" and lean towards chaos due to the ensemble disagreement reward term, such that the constellations showcased in Fig. 6 (snapshots for the absolute relational $\phi$) and Fig. S3 are not necessarily preserved.

We also evaluate the success rates for zero-shot downstream task generalization using the models trained in free-play runs with relational (R) $\phi$ and present them in Table S1. We find the performance in this case to be comparable to the absolute relational (AR) $\phi$ case. Note that in comparison to previous work [12], we use controller mode "best", not "sum" and a shorter planning horizon of 20 timesteps instead of 30. The environment is also intialized with 6 objects instead of 4. That's why the reported success rates differ from previous work. We also include the success rates for CEE-US with mode "sum", and see that using mode "best" for free play also results in improved performance in assembly tasks.

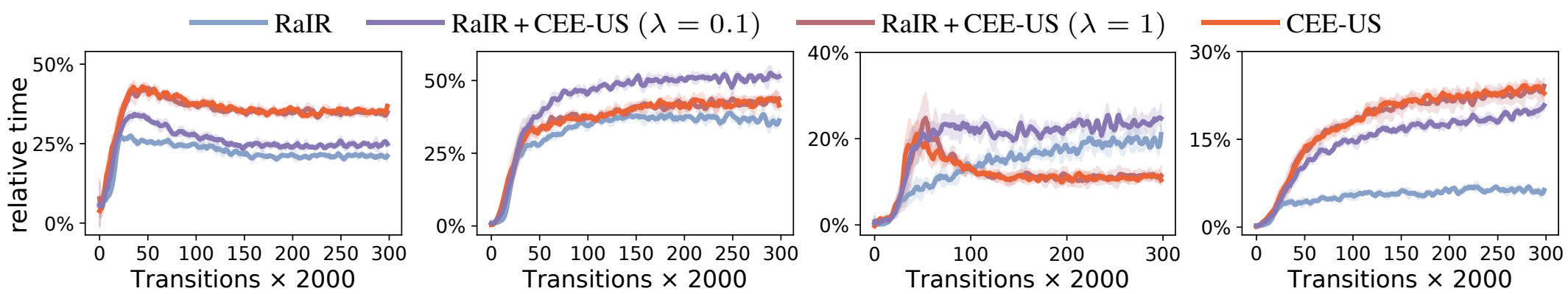

(a) 1 object moves (b) 2 or more objects move (c) object(s) in air (d) object(s) flipped

**Figure S2**: **Comparison of interactions during free play in CONSTRUCTION when combining ensemble disagreement with RaIR for different augmentation weights $\lambda$ with relational $\phi$.** Interaction metrics of free-play exploration count the relative amount of time steps spent in moving one object (a), moving two and more objects (b), moving objects in the air (c), and flipping object(s) (d). We used 5 independent seeds.

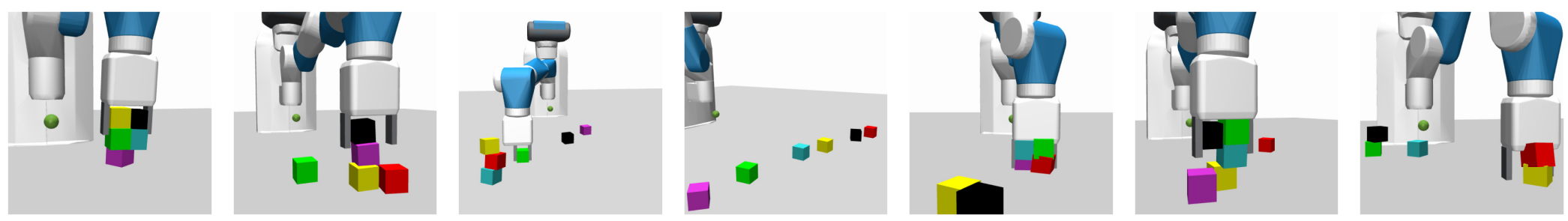

Iteration 245 Iteration 246 Iteration 247 Iteration 248 Iteration 270 Iteration 275 Iteration 295

**Figure S3**: **Snapshots from free play with RaIR + CEE-US and relational $\phi$.** We showcase snapshots of lowest entropy from exemplary rollouts at different iterations of free play. Following the regularity objective, stacks and alignments are generated. These snapshots come from a run with $\lambda = 0.1$.

**Table S1**: **Zero-shot downstream task generalization performance of RaIR + CEE-US for different $\phi$ and** $\lambda$ for assembly tasks as well as the generic pick & place task and the more chaos-oriented throwing and flipping. Results are shown for five independent seeds. AR: Absolute relative $\phi$, R: Relative $\phi$. In the row CEE-US (sum), we report success rates when we use the controller mode *sum* instead of *best*, which is the default for CONSTRUCTION in this work. In the bottom row, we report the success rates achieved via planning with ground truth models. This is to provide a baseline for how hard the task is to solve with finite-horizon planning and potentially suboptimally designed task rewards.

| | Singletower 3 | Multitower 2+2 | Pyramid 5 | Pyramid 6 | Pick&Place 6 | Throw 4 | Flip 4 |
|---|---|---|---|---|---|---|---|
| RaIR + CEE-US (R) | $\mathbf{0.80 \pm 0.07}$ | $\mathbf{0.77 \pm 0.03}$ | $\mathbf{0.47 \pm 0.04}$ | $\mathbf{0.17 \pm 0.05}$ | $\mathbf{0.90 \pm 0.01}$ | $0.38 \pm 0.02$ | $0.63 \pm 0.05$ |
| RaIR + CEE-US (AR) | $\mathbf{0.75 \pm 0.07}$ | $\mathbf{0.77 \pm 0.06}$ | $\mathbf{0.49 \pm 0.06}$ | $\mathbf{0.18 \pm 0.04}$ | $\mathbf{0.90 \pm 0.02}$ | $0.32 \pm 0.02$ | $0.63 \pm 0.08$ |
| RaIR (AR) ($\lambda = 0$) | $0.64 \pm 0.03$ | $0.62 \pm 0.03$ | $0.25 \pm 0.05$ | $0.10 \pm 0.02$ | $0.74 \pm 0.05$ | $0.21 \pm 0.01$ | $0.65 \pm 0.1$ |
| CEE-US | $0.40 \pm 0.12$ | $0.52 \pm 0.05$ | $0.14 \pm 0.09$ | $0.02 \pm 0.01$ | $\mathbf{0.90 \pm 0.02}$ | $\mathbf{0.49 \pm 0.05}$ | $0.73 \pm 0.1$ |
| CEE-US (sum) | $0.26 \pm 0.11$ | $0.48 \pm 0.13$ | $0.10 \pm 0.10$ | $0.0 \pm 0.0$ | $\mathbf{0.91 \pm 0.02}$ | $\mathbf{0.51 \pm 0.04}$ | $0.76 \pm 0.15$ |
| GT | 0.99 | 0.97 | 0.82 | 0.81 | 0.99 | 0.97 | 1.0 |

## C Experiment Results for Free Play with pure RaIR

In this section, we further discuss the zero-shot downstream task generalization performance for free play with pure RaIR and the role of the information-gain term in our intrinsic reward combination used in free play, as specified in Eq. 6. As discussed in Sec. 3.2, adding ensemble disagreement to our regularity objective leads to 1) more interaction-rich free play and 2) more robust world models which yield higher success rates for zero-shot downstream task generalization. For both the absolute relational case presented in Fig. 5 and the relational case in Fig. S2, RaIR with no disagreement term yields less interactions in terms of object(s) being moved, being in air and being flipped. This is because the exploration problem is not solved by RaIR alone. When we use ensemble disagreement as an intrinsic reward, the discovery of different types of interactions is accelerated. When one of the models in the ensemble learns a new type of dynamics, such as an object moving, the ensemble disagreement goes up, incentivizing the agent to repeat this behavior until it is learned by all models such that disagreement goes down. In the case of RaIR, this only happens implicitly: with some

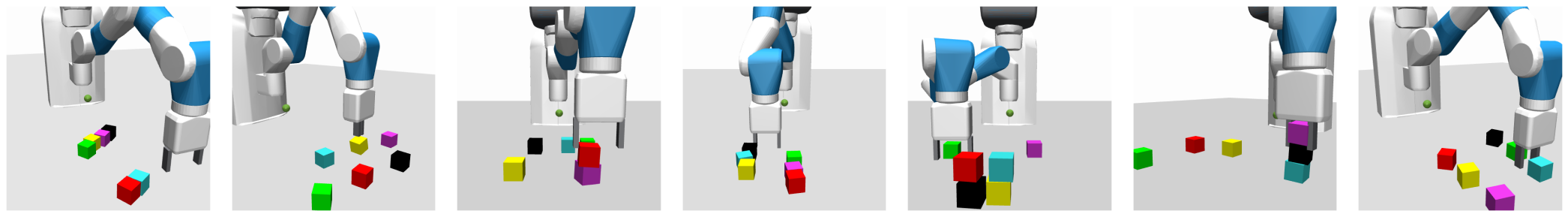

**Figure S4**: **Snapshots from free play with pure RaIR.** We showcase snapshots of lowest entropy from exemplary rollouts at different iterations of free play. These snapshots come from a run with absolute relational $\phi$.

models in the ensemble learning a certain type of dynamics in the environment, during planning, the models can hallucinate objects being aligned and creating a regular pattern with high RaIR such that these actions are executed by the controller. These false attempts also help exploration.

As the models produce better predictions, especially after free play iteration 200, we observe that more stable patterns are generated with RaIR compared to RaIR + CEE-US (Fig. 7) and the amount of time objects are moving starts decreasing Fig. 5. This is because in this case when the models get better and hallucinate less, there is no reason to leave local optima such as a spaced line unless a pattern that yields a higher regularity value can be found within the planning horizon.

The challenge of exploration with pure RaIR is also reflected in the interaction time for object(s) in air. Starting to create regular patterns such as stacks takes longer, as exploring to lift objects happens later without the disagreement reward. This is also connected to the step-wise landscape of RaIR as discussed in Sec. 3.2 such that explicit exploration via ensemble disagreement is beneficial.

As showcased in Fig. S4, we still observe the stable generation of patterns such as spaced lines later on in training as well, as these are local optima of RaIR. However, we start to see more stacks generated in the later stages of free play. In Fig. 7, the highest RaIR value achieved for the different variants are showcased throughout training. Pure RaIR, achieves slightly higher regularity then RaIR + CEE-US. This is also because pure RaIR, tends to generate more regular patterns that feature all objects, i.e. all objects are in-line or build a square. With RaIR + CEE-US, as some chaotic behavior is injected to free play via ensemble disagreement, more local regularities such as a stack of 2, with the rest of the objects in disorder, are likely to emerge.

Through injecting ensemble disagreement into free play, the robustness of the learned world models is also increased as they are guided by their own epistemic uncertainty [12]. During free play, data is actively collected from regions where the models are uncertain, acting as their own adversary. This in turn makes the models more robust for deployment in model-based planning in the follow-up extrinsic phase, where the accuracy of model predictions is paramount for good performance. This is reflected in the downstream task performance evaluations in Fig. 8, where RaIR + CEE-US consistently outperforms both RaIR and CEE-US in the assembly tasks. Note that as regularity explicitly favours alignments such as stacks, unlike CEE-US, these dynamics are explored better, leading to higher success rates. This also showcases the importance of guiding free play towards regularity. In Fig. S5, the results for the pick & place, throwing and flipping tasks are shown. Due to the increased robustness of the model with the disagreement term, we indeed observe better performance for RaIR + CEE-US and CEE-US for the Pick & Place task compared to pure RaIR. This is also true for the throwing task. However, another contributing factor here is that models with disagreement favor more chaotic behaviors and perform more “throwing”-like behaviors during free play. As CEE-US has no bias towards regularity, it performs best, whereas pure RaIR performs worse than RaIR + CEE-US. Interestingly, for the flipping 4 objects task we found performance for RaIR and RaIR + CEE-US to be comparable despite the significantly reduced amount of time spent flipping objects in the case of pure RaIR, as can be seen in Fig. 5. Upon inspecting the data generated during free-play, we hypothesize this is because unlike RaIR + CEE-US, which flips and rolls objects together in a chaotic way, we found RaIR to produce more isolated flipping of individual objects.

## D Experiment Results for Re-creating Existing Patterns

As presented in Sec. 3.4, we test whether we can re-create existing regularities in the environment by simply optimizing for RaIR with iCEM, using ground truth models. As for the pyramid with

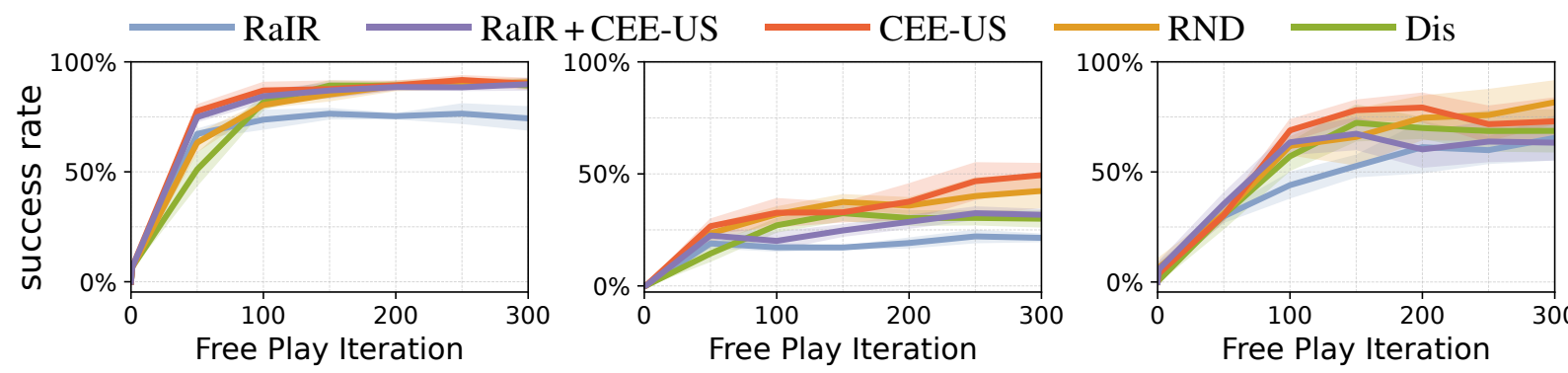

**Figure S5**: **Downstream Task Performance for Pick & Place and the more chaotic tasks of throwing and flipping** with only RaIR ($\lambda = 0$), RaIR + CEE-US ($\lambda = 0.1$), CEE-US, RND and Disagreement. We use absolute relational $\phi$ for RaIR computations. Results are shown for 5 independent seeds.

3 objects in Sec. 3.4, we initialize different regular structures outside of the robot's manipulability range and test whether these regular patterns can be re-created, merely by maximizing for RaIR. We test for the re-creation of a singletower with 3 and 4 objects, 2 towers with 2 objects each (referred to as multitower 2+2), as well as a spaced line and a rhombus with 4 objects. We test this with ground truth models for 15 independent rollouts for each structure and report the re-creation rates. Example rollouts are illustrated in Fig. S6. Note that due to the limited sample-budget with iCEM and the finite-horizon, we don't necessarily converge to the global minima, which corresponds to the full recreation of the structure. However, in all of the tested cases, the generated structures repeat at least one prominent sub-structure present in the underlying regular constellation by optimizing for RaIR.

For *Singletower 3*, the entire stack of 3 gets recreated 73% of the time. A partial recreation with a stack of 2 blocks is observed in all but one of the remaining cases.

When a *Singletower 4* is initialized outside of the robot's range, the full tower with 4 blocks gets recreated 40% of the time. In the remaining cases, either a tower of 3 (33%) or towers of 2 (27%) are built.

For the challenging *Multitower 2+2* case, the two towers are built, with the same distance to each other as in the original pattern, 20% of the time. An example of this "complete" recreation is illustrated in Fig. S6c. Otherwise, 53% of the time a stack of 2 is built (Fig. S6d) or a spaced line repeating the relative position of the two towers in the original pattern.

For the patterns on the ground, namely *Spaced Line* and *Rhombus*, the recreation rates are higher since the exploration problem is less prominent. At least 75% of the original pattern is re-created at each rollout, i.e. for the case of 4 objects, at least 3 objects follow the original pattern. The complete *Spaced Line* is recreated 80% and the entire rhombus 73% of the test rollouts.

In these experiments, we use RaIR with $\phi(s_i, s_j) = \{(|\lfloor s_{i,x} - s_{j,x} \rceil|, |\lfloor s_{i,y} - s_{j,y} \rceil|, |\lfloor s_{i,z} - s_{j,z} \rceil|)\}$. This is because in the case of existing structures in the scene, we don't need/want to inject any biases, e.g. towards vertical alignments, into the optimization. As the existing pattern is outside of the manipulability range of the robot, re-creating the pattern becomes a direct global optimum for RaIR, as all regularities reoccur. However, if we restrict ourselves to the $x$-$y$ subspace, this is no longer the case: even for a rhombus, a stack built with the blocks in-reach becomes the global optimum. This is because all the blocks in the stack then have the same $x$-$y$ relation to the blocks in the rhombus.

# E Experiment Results for RaIR in Colored SHAPEGRIDWORLD with GT Models

In colored SHAPEGRIDWORLD, $S_{\text{obj}}$ includes not only the $x$-$y$ positions, but also a binary variable encoding the different classes of colors. For $c$ colors, we have an encoding of length $\text{ceil}(\log_2 c)$. So when we are computing relational RaIR, we also take into account the relations between colors. Take as example the case where we have two color classes blue (0) and orange (1) and use absolute relational $\phi$: if all blue objects are in a horizontal spaced line, we have repetitions of $|[x_i + \delta_x, 0, 0] - [x_i, 0, 0]| = [\delta_x, 0, 0]$ between neighboring objects with spacing $\delta_x$. If we have a line with an alternating pattern of colors in sequence, then we again have redundancies. For instance, for neighboring entities orange-blue we have: $|[x_i + \delta_x, 0, 1] - [x_i, 0, 0]| = [\delta_x, 0, 1]$. Since we have absolute relational $\phi$, we get the same symbol for the blue-orange neighbors $|[x_i + \delta_x, 0, 0] - [x_i, 0, 1]| = [\delta_x, 0, 1]$.

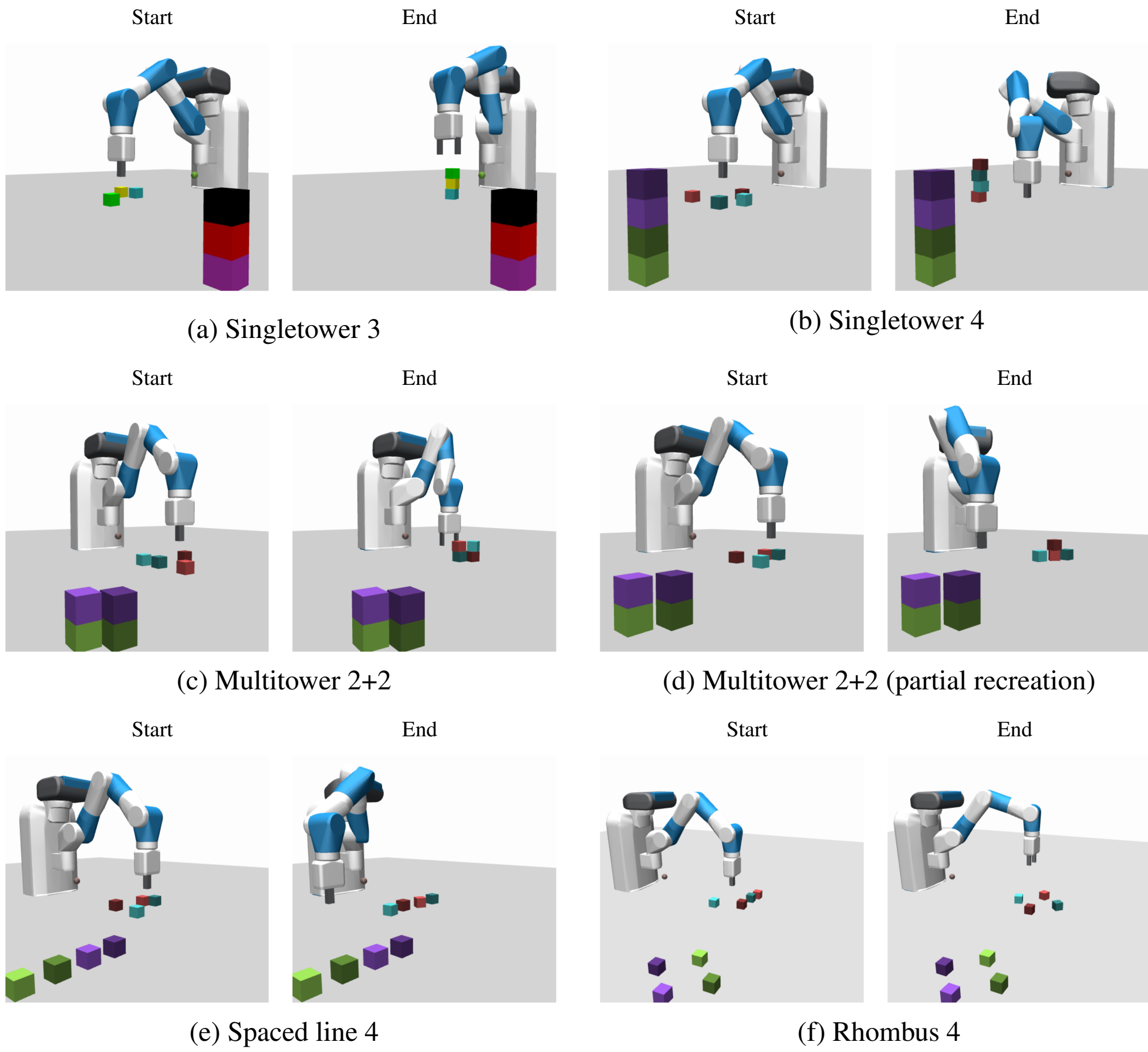

**Figure S6**: Different regular structures initialized outside of the robot's reach at the start of the episode ($t = 0$) and re-created by optimizing for RaIR with GT models. Showcased here for the end of the episode ($t = 200$).

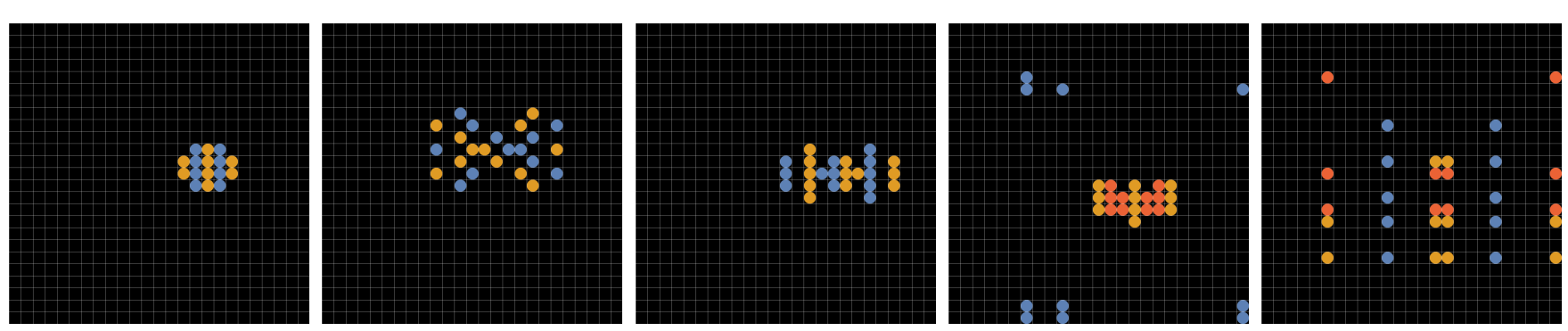

**Figure S7**: **Including color into the regularity computations in SHAPEGRIDWORLD.** We show generated patterns when maximizing for RaIR, with x-y dimensions and color as an extra symbol. In this case, RaIR tries to find patterns that are not just spatially symmetric but also regular in color.

As a result, a blue-orange-blue-orange-. . . alternating line is also an optimum for RaIR. Additional patterns generated with RaIR in colored SHAPEGRIDWORLD with 2 and 3 colors using GT models are shown in Fig. S7.

## F Experiment Results for RaIR in Custom CONSTRUCTION

We test regularity on a variant of CONSTRUCTION with diverse object shapes and masses (more details in Appendix. I.1).

### F.1 RaIR with GT models

We apply RaIR to custom CONSTRUCTION with different shapes and masses, where we compute regularity between the $x$-$y$ positions of the center of mass of objects. Here our aim is to show the generality of RaIR and that we are not constrained to identical entities. The generated constellations when optimizing for RaIR using GT models are showcased in Fig. S8.

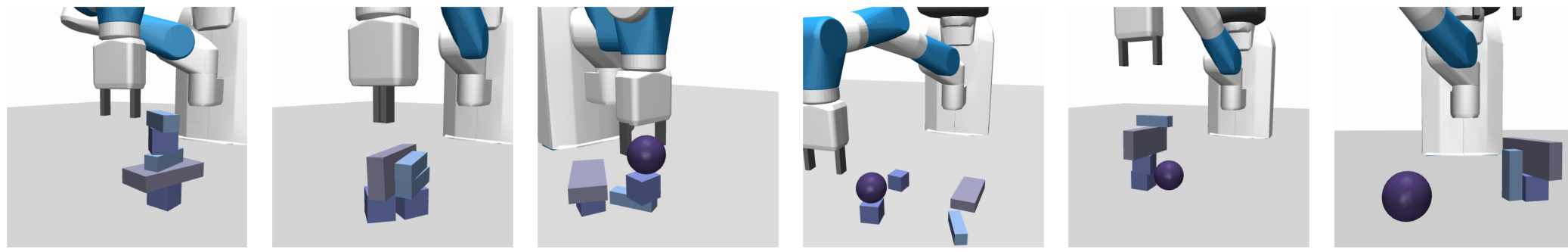

**Figure S8: Applying RaIR to a custom CONSTRUCTION environment with different shapes and masses with GT models.** We show generated arrangements when maximizing for RaIR with x-y positions of objects (center of mass) in 2 instances of the environment: 1) with 2 cubes, 2 short columns and 1 flat block (the left 2 images) and 2) with 2 cubes, 1 ball, 1 column and 1 flat block (the right 4 images).

### F.2 RaIR in Free Play with Learned Models

We run free play in the custom CONSTRUCTION. During free play, we initialize one instance from each object type such that we have 1 cube, 1 ball, 1 column and 1 flat block. We then evaluate the zero-shot downstream task performance of RaIR + CEE-US and CEE-US for 3 different stacking tasks (Fig. S9). The results are shown in Table S2, where we obtain higher success rates with RaIR + CEE-US.

The free play parameters (model and controller settings) are identical to the standard CONSTRUCTION environment given in Table S6. The only difference is, we include a categorical variable to encode the different object types and append this to the observation vector for all objects. This is done in the same fashion as the static observations proposed in Sancaktar et al. [12]. For RaIR + CEE-US, we again use $\lambda = 0.1$.

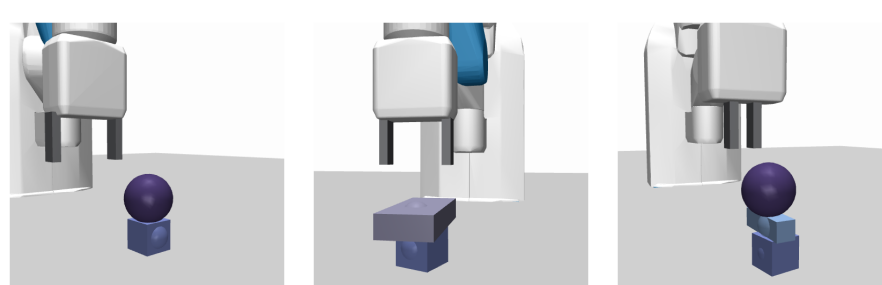

Stack Cube + Ball | Stack Cube + Flat Block | Stack Cube + Column + Ball

**Figure S9: Downstream tasks for custom CONSTRUCTION.**

**Table S2: Zero-shot downstream task generalization performance of RaIR + CEE-US for custom CONSTRUCTION.** Results are shown for five independent seeds, for models evaluated after 300 free play iterations (equivalent to 2K $\times 300 = 600$K transitions).

| | Stack Cube + Ball | Stack Cube + Flat Block | Stack Cube + Column + Ball |
|---|---|---|---|
| RaIR + CEE-US | $\mathbf{0.66 \pm 0.10}$ | $\mathbf{0.67 \pm 0.09}$ | $\mathbf{0.15 \pm 0.05}$ |
| CEE-US | $0.38 \pm 0.11$ | $0.45 \pm 0.04$ | $0.09 \pm 0.03$ |

## G Experiment Results for RaIR in the Quadruped and Walker Environments

We showcase that regularity also finds application in locomotion environments, where we compute RaIR on the world coordinates of the robot joints. Environment details for both Quadruped and Walker can be found in Appendix. I.1.

We run free play with RaIR + CEE-US ($\lambda = 0.02$) and CEE-US in the Quadruped environment. With RaIR + CEE-US, we indeed observe more regular poses during free play, as reflected in the RaIR values shown in Fig. S10. We evaluate the zero-shot downstream performance of the learned models for the Roboyoga poses shown in Fig. S11. The success rates for the models at the end of free play (Table S3), as well as their temporal evolution (Fig. S12) are reported. We evaluate success for a pose as having a distance smaller than a threshold of 0.4 to the goal pose for more than 10 timesteps. We only see marginal improvements with RaIR + CEE-US compared to CEE-US. We hypothesize that this is because model-based approaches can generalize very well in locomotion environments and CEE-US also provides sufficient exploration. It

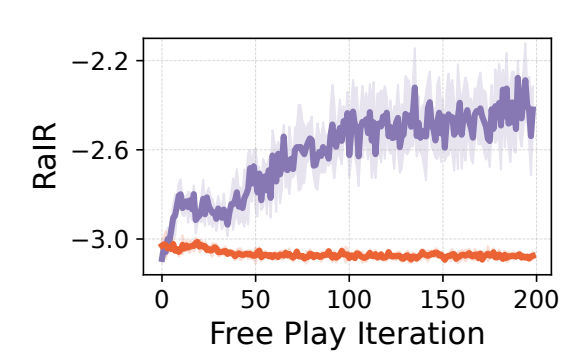

**Figure S10: Highest RaIR in Quadruped free play** with RaIR + CEE-US and CEE-US.

only comes down to whether the pose can be held longer stably, where in the balancing poses RaIR + CEE-US has a slight edge over the course of free play (especially for *Balance Front*). For any goal that is shown in Fig. S11 and not included in Table S3 (i.e. the poses with Quadruped lying on its back), we get perfect success rates for both RaIR + CEE-US and CEE-US. We didn't run free play in the Walker environment, but we expect model-based approaches to be very strong there altogether, as the environment dynamics are very simple.

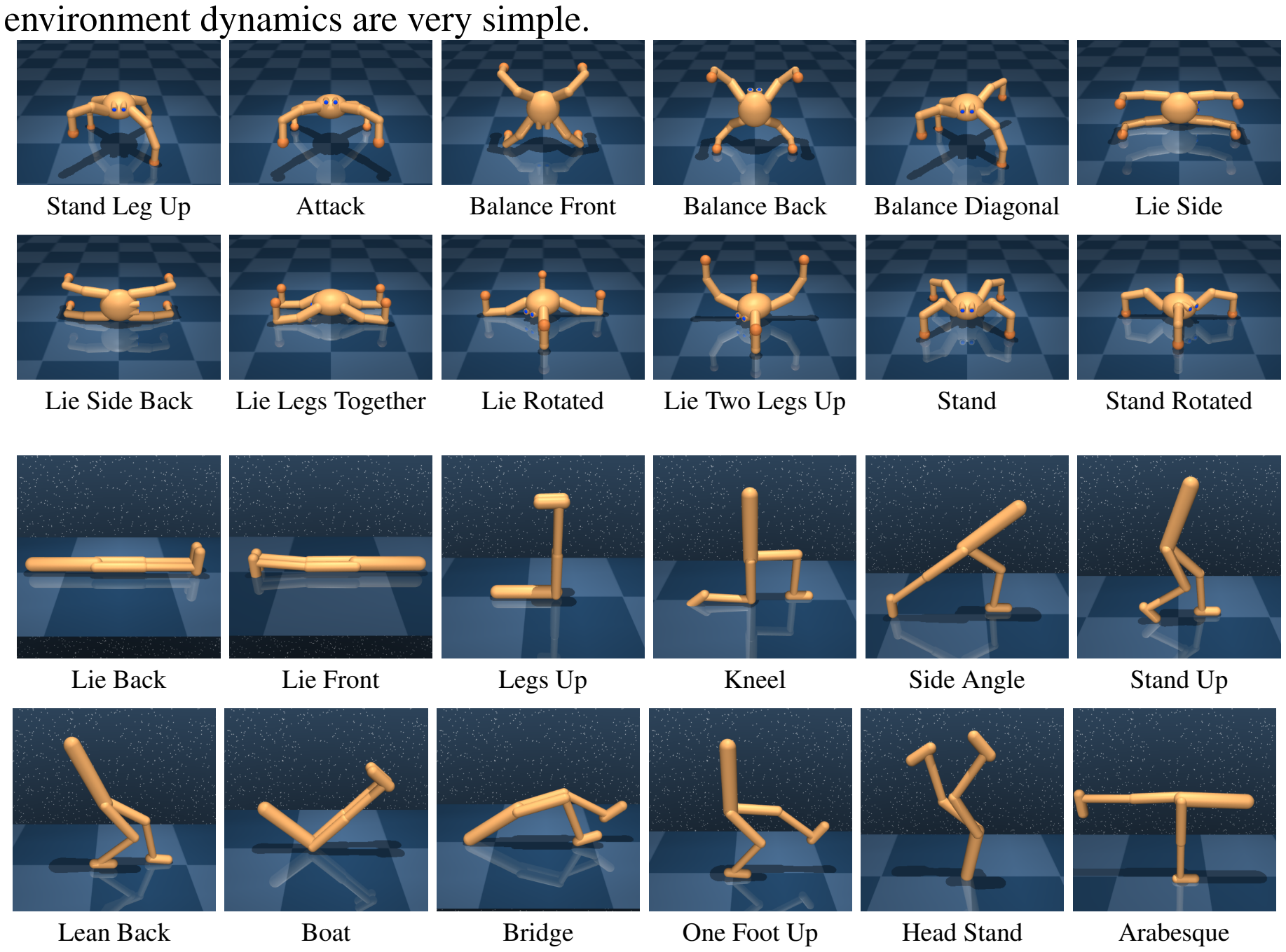

**Figure S11**: **Goals in the RoboYoga benchmark** as proposed in Mendonca et al. [25] for Quadruped (top 2 rows) and Walker environments (bottom 2 rows).

# H CEE-US

In this section, we present the details of CEE-US [12], which we build upon in this work. CEE-US uses structured world models together with model-based planning during exploration, achieving increased sample-efficiency and superior downstream task performance compared to other intrinsically-motivated RL baselines. The free-play pseudocode is presented in Alg. S1. This free-play structure is used for all methods presented in our paper by swapping out the intrinsic reward term (line 5)

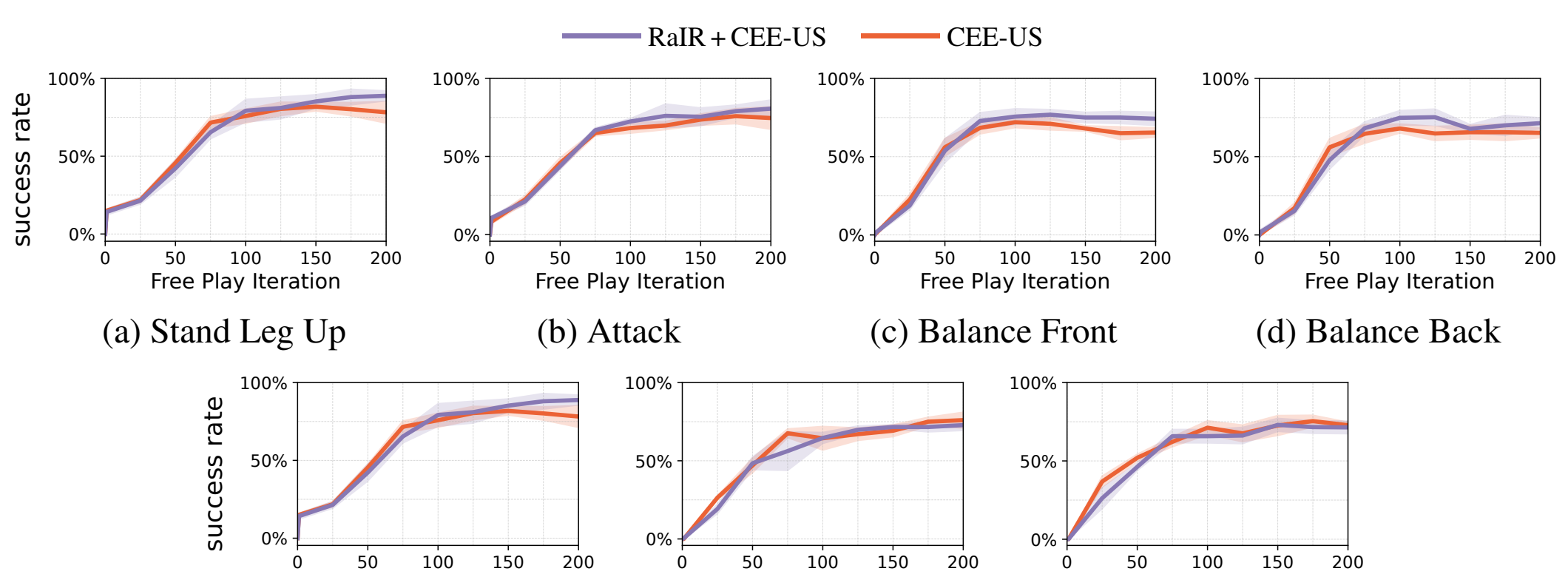

(a) Stand Leg Up (b) Attack (c) Balance Front (d) Balance Back

(e) Balance Diagonal (f) Lie Side (g) Lie Side Back

**Figure S12**: **Downstream task performance for a subset of Quadruped RoboYoga tasks** with RaIR + CEE-US ($\lambda = 0.02$) and CEE-US, evaluated for models checkpointed throughout free play. We use absolute relational $\phi$ for RaIR computations. (Five independent seeds)

**Table S3**: **Zero-shot downstream task generalization performance of RaIR + CEE-US and CEE-US for the Quadruped RoboYoga benchmark.** Results are shown for five independent seeds, for models evaluated after 200 free play iterations (equivalent to 2K $\times$ 200 $=$ 400K transitions).

| | Stand Leg Up | Attack | Balance Front | Balance Back | Balance Diagonal | Lie Side | Lie Side Back | Stand | Stand Rotated |
|---|---|---|---|---|---|---|---|---|---|
| RaIR + CEE-US | $0.89 \pm 0.03$ | $0.81 \pm 0.06$ | $0.74 \pm 0.04$ | $0.71 \pm 0.03$ | $0.89 \pm 0.03$ | $0.73 \pm 0.03$ | $0.71 \pm 0.04$ | $0.89 \pm 0.04$ | $0.85 \pm 0.06$ |
| CEE-US | $0.78 \pm 0.07$ | $0.75 \pm 0.07$ | $0.65 \pm 0.03$ | $0.65 \pm 0.03$ | $0.78 \pm 0.07$ | $0.76 \pm 0.05$ | $0.73 \pm 0.02$ | $0.78 \pm 0.07$ | $0.80 \pm 0.06$ |

with only ensemble disagreement (CEE-US), combination of our regularity objective with ensemble disagreement (RaIR + CEE-US) or pure regularity (RaIR), as well as the other baselines RND and Dis (see Sec. J).

**Algorithm S1** **Free Play in Intrinsic Phase (taken from [12])**

**Input:** $\{(\tilde{f}_{\theta_m})_{m=1}^{M}\}$: Randomly initialized ensemble of GNNs with $M$ members, $D$: empty dataset, `Planner`: iCEM planner with horizon $H$
**while** explore **do** ▷ *Explore with MPC and intrinsic reward*
  **for** $e = 1$ **to** `num_episodes` **do**
    **for** $t = 1$ **to** $T$ **do** ▷ *Plan to maximize intrinsic reward*
      $a_t \leftarrow \texttt{Planner}(s_t, \{(\tilde{f}_{\theta_m})_{m=1}^{M}\}, r_{\text{intrinsic}})$ ▷ *e.g. RaIR with disagreement Eq. 6*
      $s_{t+1} \leftarrow \texttt{env.step}(s_t, a_t)$
    $\mathcal{D} \leftarrow \mathcal{D} \cup \{(s_t, a_t, s_{t+1})_{t=1}^{T}\}$
  **for** $l = 1$ **to** $L$ **do** ▷ *Train models on dataset for L epochs*
    $\theta_m \leftarrow$ optimize $\theta_m$ using $\mathcal{L}_m$ on $\mathcal{D}$ for $m = 1, \ldots, M$
**return** $\{(\tilde{f}_{\theta_m})_{m=1}^{M}\}, \mathcal{D}$

### H.1 GNN Architectural Details

Message-passing Graph Neural Networks (GNN) are deployed as world models. The same GNN architecture is used as in CEE-US [12]. For these structured world models, we consider object-factorized state spaces with $\mathcal{S} = (\mathcal{S}_{\text{obj}})^N \times \mathcal{S}_{\text{robot}}$. Each node in the GNN corresponds to an object and the robot/actuated agent is differentiated from the object nodes as a global node. The concatenation of the robots's state $s_t^{\text{robot}}$ and the action $a_t$ is represented as a global context $c = [s_t^{\text{robot}}, a_t]$. We have a fully-connected GNN. The node update function $g_{\text{node}}$ and the edge update function $g_{\text{edge}}$ model the dynamics of the entities/objects, and their pairwise interactions respectively. These functions are both Multilayer Perceptrons (MLP). In the following, we denote the state of the $i$-th object $s_{t,\text{obj}_i}$ at timestep $t$ as $s_t^i$ for simplicity. The object node attributes in the GNN are updated as:

$$e_t^{(i,j)} = g_{\text{edge}}\big([s_t^i, s_t^j, c]\big) \tag{S1}$$

$$\tilde{s}_{t+1}^i = g_{\text{node}}\big([s_t^i, c, \text{aggr}_{i \neq j}\, (e_t^{(i,j)})]\big). \tag{S2}$$

where $[\cdot, \ldots]$ denotes concatenation, $e_t^{(i,j)}$ is the edge attribute between two neighboring nodes $(i, j)$. For the permutation-invariant aggregation function given by aggr, we use the mean.

The robot state, which is treated as a global node, is computed using the global aggregation of all edges with a separate global node MLP $g_{\text{global}}$:

$$\tilde{s}_{t+1}^{\text{robot}} = g_{\text{global}}\big([c, \text{aggr}_{i,j}\, (e_t^{(i,j)})]\big). \tag{S3}$$

Moreover, the GNN predicts the changes in the dynamics such that $\tilde{s}_{t+1} = s_t + \text{GNN}(s_t, a_t)$.

### H.2 Planning Details

For planning, we use the improved Cross-Entropy Method (iCEM) [19]. The planner minimizes the cost, corresponding to negative reward $c(s_t, a_t, s_{t+1}) = -r(s_t, a_t, s_{t+1})$, where $r$ can be intrinsic rewards $r_{\text{intrinsic}}$ or extrinsic task rewards $r_{\text{task}}$. The extrinsic task rewards are assumed to be given by the environment.

At each timestep $t$ in the environment, the planner samples $P$ action sequences, each with length $H$, i.e. the planning horizon. These actions are rolled out either in the ground truth model (perfect simulations) or in the imagination of a learned model (imperfect simulations), generating corresponding $P$ state sequences with length $H$. In order to assign a cost to each of the $P$ trajectories, we need to aggregate the cost over the horizon $H$. A typical choice here is `sum`, where the cost over the length of the trajectory is simply summed up: $\texttt{cost}^{(p)} = \sum_{h=0}^{H-1} c(s_{t+h}^{(p)}, a_{t+h}^{(p)}, s_{t+h+1}^{(p)})$.

However, this type of aggregation is not suitable for cases where a decrease in cost can in general be preceded by an initial increase. In these cases, using the mode `best`, that assigns the plan $p$ the cost of the "best" timestep over the planning horizon with $\texttt{cost}^{(p)} = \min\Big(\{c(s_{t+h}^{(p)}, a_{t+h}^{(p)}, s_{t+h+1}^{(p)})\}_{h=0}^{H-1}\Big)$ is a better suited choice. We also empirically found this controller mode to be better at picking up *sparse* signals. What we mean here is that, in the example of stacking, it is hard to find a sampled trajectory that stacks the objects in a stable way with a limited sample-budget as this poses an exploration challenge. However, if we manage to find an action sequence that brings the cubes on top of each other, albeit in an unstable way, favoring this solution with `best` and keeping this solution in the elite set is beneficial. This can be explained as follows: In iCEM the $K$ plans with the lowest assigned cost are chosen to be the elite set, which is then used to fit the sampling distribution of iCEM. As a fraction $\xi$ of these elites is potentially shifted to the next internal iCEM iteration (`keep_elites`), and possibly to the next timestep (`shift_elites`), keeping these solutions that "fail" and yet bring us closer to the actual solution provides a better strategy to solve tasks which pose an exploration challenge such as stacking. Here, we are also relying on the fact that we are re-running optimization every timestep $t$ in the environment with online model predictive control, such that we have the opportunity to correct these initially "wrong" solutions and find their "stable" counterparts. Note that this mode of the controller is a more unstable mode compared to `sum`. Especially with imperfect world models, where the model can hallucinate as the model errors accumulate over the planning horizon, mode `best` can pick up these falsely imagined future states with low cost. It also doesn't account for the fact that the planned trajectory keeps the lowest cost over multiple timesteps, such that a trajectory where an object flies through the goal location for a single timestep has the same cost as a trajectory where the object lands in the goal position and stays there. To account for this, we use $\texttt{cost}^{(p)} = \min\Big(\{c(s_{t+h}^{(p)}, a_{t+h}^{(p)}, s_{t+h+1}^{(p)})\}_{h=1}^{H-1}\Big)$, where we don't take into account the first timestep of the plan with $h = 0$. Although this is not a robust solution, we found it to empirically work well. Quantitatively, stacking 3 objects when planning with ground truth models yields 99% success rate for mode `best`, whereas only 47 % success rate for `sum`, using the same reward function in both cases. Even in the case of perfect dynamics predictions with GT models, this showcases the importance of the controller mode to be able solve tasks with sparse reward signals. This is further showcased in Table S1, when comparing the results for CEE-US with default `best` and `sum`.

# I Experiment Details

In this section, we provide experimental details and hyperparameter settings.

## I.1 Environment Details

**SHAPEGRIDWORLD** This is a discrete 2D grid, where each entity/agent is controlled individually in the $x$-$y$ directions. This means entities are controlled one at a time and actuation keeps iterating over the entities, where we use an object persistency of 10 timesteps. The action $a_t$ is 2 dimensional, controlling the agent in $x$-$y$ directions separately and is applied on the current actuated entity $i$ in the grid. As we are operating in a discrete grid, the actions are actually discrete such that the agent can move one grid cell to the left/right and up/right (if the target grid cell is not occupied) or stay at the current grid cell. In order to make this environment work with the default iCEM implementation with a Gaussian sampling distribution, we perform a discretization step before inputting the sampled actions to the environment. For the experiment results with GT models presented in Fig. 1, we use a grid size of $25 \times 25$ with 16 and 32 objects. In colored SHAPEGRIDWORLD with $c$ colors, each object is assigned one of the $c$ available color options, and the observation for each entity also includes binary encoding for color with length $\mathrm{ceil}(\log_2 c)$.

**CONSTRUCTION** This is a multi-object manipulation environment as an extension of the Fetch Pick & Place environment proposed in [24]. We also applied the two modifications from Sancaktar et al. [12]. 1) The table in front of the robot is replaced with a large plane such that objects cannot fall off during free play, but can still be thrown/pushed outside of the robot's reach. 2) In Li et al. [24], the object state also contained the object's position relative to the gripper which was removed, as it already introduces a relational bias in the raw state representation. Details on the dimensionalities of the object and robot state spaces can be found in Table S5.

**Custom CONSTRUCTION** As an extension of the standard CONSTRUCTION, there are 4 different object types with different masses: cube (same as in original CONSTRUCTION) with size 5cm and mass 2 (mass in the default unit of Mujoco), flat block with size $16 \times 7 \times 3$cm and mass 1.5, column (upright block) with size $3 \times 3 \times 10$cm and mass 1, and a ball with diameter 8cm and mass 1. We also include categorical variables for each object type as static observations. During free play, we initialize one of each object type, so we have 4 objects. Otherwise the specifications are the same as in Table S5.

**Quadruped & Walker** These environments are taken from the DeepMind Control Suite [45]. To be able to perform RaIR computations, we modify the observation space of each environment and append the world coordinates of the robot's joints to the observations vector. In Quadruped, we take the toe and knee positions. We used $x$-$y$-$z$ coordinates with the GT models (results shown in Fig. 10), and $x$-$y$ for free play in Quadruped with learned models. In principle, we can also include the ankle and hip joints, but we observed similar performance in this case. For Walker, we append the feet, leg and torso $x$-$y$ positions.

### I.2 Parameters for Ground Truth Model Experiments with RaIR

The controller parameters used when optimizing RaIR with ground truth (GT) models are given in Table S4. To compute RaIR, we perform a discretization step in the CONSTRUCTION environment as it is continuous. For GT models, that produce perfect mental simulations, we can choose a small bin size of 1cm. In comparison, the size of one block in the environment is 5 cm. The bin size also gives us the upper bound of the *regularity precision* that can be achieved during optimization, e.g. a perfectly aligned stack vs. a zigzagged stack. Note however that the higher the precision is, the harder it typically gets for the controller to converge to global optima with a horizon of 30 timesteps and a limited sample-budget.

As discussed in Sec. A, this is also a constraint when we are computing RaIR in the $x$-$y$-$z$ subspace. Due to this, for the re-creation of existing patterns experiments presented in Sec. 3.4 and Sec. D, we compute RaIR with a bin size of 2.5cm and increase the number of sampled trajectories $P$ to 512. Although we could further increase the bin size, we choose this value to not negatively impact the precision of the re-created structures. In custom CONSTRUCTION, we also use a bin size of 2.5cm, when optimizing RaIR with GT models.

For Quadruped, we also use a bin size of 5cm. For the GT model experiments, we compute RaIR over the $x$-$y$(-$z$) positions of the Quadruped's toe and knee joints. We didn't see significant qualitative differences when including the $z$-positions in the regularity computation. For Walker, we use a bin size of 1cm and compute RaIR over the $x$-$y$ positions of the Walker's feet, legs and torso.

### I.3 Free Play with Learned Models

The environment properties with the episode lengths and model training frequencies are given in Table S5. Six objects are present in CONSTRUCTION during free play. The parameters for the GNN model architecture as well as the training parameters for model learning are listed in Table S6. For the RaIR computations in free play, we use a bin size of 5cm, which is equivalent to the size of a block.

For custom CONSTRUCTION, 4 objects (one of each object type) are present during free play. The hyperparameters are the same as for standard CONSTRUCTION, and a bin size of 5cm is used.

The set of the hyperparameters for the iCEM controller used in the intrinsic phase of RaIR + CEE-US, RaIR and CEE-US are the same as presented in Table S4. The only difference to the GT model case

**Table S4**: Base settings for iCEM. These hyperparameters are used when using GT models to optimize RaIR.

(a) General settings.

| **Parameter** | **Value** |
|---|---|
| Number of samples $P$ | 128 |
| Horizon $H$ | 30 |
| Size of elite-set $K$ | 10 |
| Colored-noise exponent $\beta$ | 3.5 |
| *CEM-iterations* | 3 |
| Noise strength $\sigma_{\text{init}}$ | 0.8 |
| Momentum $\alpha$ | 0.1 |
| `use_mean_actions` | Yes |
| `shift_elites` | Yes |
| `keep_elites` | Yes |
| Fraction of elites reused $\xi$ | 0.3 |
| Cost along trajectory | `best` |

(b) Environment-specific settings.

| SHAPEGRIDWORLD | |
|---|---|
| **Parameter** | **Value** |
| Number of samples $P$ | 64 |

| CONSTRUCTION | |
|---|---|
| **Parameter** | **Value** |
| Same as general settings | |

| Quadruped & Walker | |
|---|---|
| **Parameter** | **Value** |
| Number of samples $P$ | 64 |
| Colored-noise exponent $\beta$ | 2.5 |
| Noise strength $\sigma_{\text{init}}$ | 0.3 |
| Cost along trajectory | `sum` |

**Table S5**: Environment settings for CONSTRUCTION. 2000 transitions (20 episodes with 100 timesteps each) are generated within one training iteration of free play.

| CONSTRUCTION | |
|---|---|
| **Parameter** | **Value** |
| Episode Length | 100 |
| Train Model Every | 20 Episodes |
| Action Dim. | 4 |
| Robot/Agent State Dim. | 10 |
| Object Dynamic State Dim. | 12 |
| Num. of Objects During Free Play | 6 |

**Table S6**: Settings for GNN model training in intrinsic phase of RaIR + CEE-US and CEE-US. (Same as in [12]) These settings are used for both CONSTRUCTION and Custom CONSTRUCTION free play runs.

| **Parameter** | **Value** |
|---|---|
| Network Size of $g_{\text{node}}$ | $2 \times 128$ |
| Network Size of $g_{\text{edge}}$ | $2 \times 128$ |
| Network Size of $g_{\text{global}}$ | $2 \times 128$ |
| Activation function | ReLU |
| Layer Normalization | Yes |
| Number of Message-Passing | 1 |
| Ensemble Size | 5 |
| Optimizer | ADAM |
| Batch Size | 125 |
| Epochs | 25 |
| Learning Rate | $10^{-5}$ |
| Weight Decay | 0.001 |
| Weight Initialization | Truncated Normal |
| Normalize Input | Yes |
| Normalize Output | Yes |
| Predict Delta | Yes |

**Table S7**: Settings for the MLP model training in intrinsic phase of RaIR + CEE-US and CEE-US in the Quadruped free play runs.

| Parameter | Value |
|---|---|
| Episode Length | 100 |
| Train Model Every | 20 Episodes |
| Network Size | $3 \times 600$ |
| Activation function | SiLU |
| Ensemble Size | 5 |
| Optimizer | ADAM |
| Batch Size | 128 |
| Epochs | 25 |
| Learning Rate | 0.0001 |
| Weight decay | 0.0001 |
| Weight Initialization | Truncated Normal |
| Normalize Input | Yes |
| Normalize Output | Yes |
| Predict Delta | Yes |

is, we use a planning horizon of 20 timesteps for free play. We also use the exact same settings for the RND baseline. For Disagreement, we have a planning horizon of 1 during free play.

For the Quadruped environment, we use an MLP ensemble instead of a GNN ensemble. The hyperparameters are reported in Table S7. The iCEM controller parameters used in free play for Quadruped are the same as the ones for GT models in Table S4, except for the noise strength $\sigma_{\text{init}} = 0.5$.

### I.4 Extrinsic Phase: Zero-shot Downstream Task Generalization

In this section, we provide details on the extrinsic phase following free play, where the learned GNN ensemble is used to solve downstream tasks zero-shot via model-based planning.

#### I.4.1 Details on Downstream Tasks and Reward Functions

The reward functions for all the CONSTRUCTION downstream tasks are computed as specified in Sancaktar et al. [12], where for all the assembly tasks we use the same structure as in the stacking reward. However, we do one modification to the original reward computation in the assembly tasks. The assembly task reward is sparse incremental with reward shaping, where the reward also contains the distance between the gripper and the position of the next block to be stacked. We modify how the next block ID is computed in the original implementation from Sancaktar et al. [12]. Instead of naively checking the number of unsolved objects to obtain the next block ID irrespective of order, we determine the next block to be the next unsolved block in the order. We found this modification to be important especially for the Pyramid tasks, where the sub-optimal next block computation might lead to the agent receiving a reward to be close to the wrong block, in the case the robot places blocks with $i >$ `next_block_id` to their goal locations with just the sparse reward.

For the RoboYoga experiments, the reward function is the same as proposed in [25] (taken from their code in `https://github.com/orybkin/lexa-benchmark`). It is a dense reward as the shortest angle distance to the goal pose.

#### I.4.2 Planning Details for Downstream Tasks

The controller settings for the different downstream tasks are shown in Table S8, which are the same settings used in [12].

**Table S8**: Settings for the iCEM controller used for zero-shot generalization in the extrinsic phase of RaIR + CEE-US and CEE-US. The settings not specified here are the same as the general settings given in Table S4. The settings are exactly the same as in [12].

| Task | Controller Parameters | | | | |
|---|---|---|---|---|---|
| | Horizon $h$ | Colored-noise exponent $\beta$ | `use_mean_actions` | Noise strength $\sigma_{\text{init}}$ | Cost Along Trajectory |
| CONSTRUCTION-Stacking | 30 | 3.5 | No | 0.5 | `best` |
| CONSTRUCTION-Pick & Place | 30 | 3.5 | Yes | 0.5 | `best` |
| CONSTRUCTION-Throwing | 35 | 2.0 | Yes | 0.5 | `sum` |
| CONSTRUCTION-Flipping | 30 | 3.5 | No | 0.5 | `sum` |

**Table S9**: Settings for the RND predictor $f_\theta$ and target $\hat{f}$ networks.

| Parameter | Value |
|---|---|
| Train Predictor Every | 2K Transitions |
| Network Size | $1 \times 256$ |
| Embedding Dim $k$ | 128 |
| Activation function | ReLU |
| Optimizer | ADAM |
| Batch Size | 256 |
| Epochs | 10 |
| Learning Rate | $10^{-4}$ |
| Weight Initialization | Orthogonal |
| Normalize Input | Yes |

# J Details on Baselines

## J.1 Random Network Distillation

Random network distillation (RND) [21] has two networks: a predictor network and a randomly-initialized and fixed target network. The predictor network $f_\theta : S \to \mathbb{R}^k$ is trained on the data collected by the agent to match the outputs of the target network $\hat{f} : S \to \mathbb{R}^k$, where $k$ is the embedding dimension. The error between the predictor and target networks for the state $s_t$ is used as the intrinsic reward signal:

$$r_{\text{intrinsic}} = \|f_\theta(s_t) - \hat{f}(s_t)\|^2, \tag{S4}$$

where the idea is that this error will be higher for novel states. RND can be seen as an extension of count-based methods to continuous domains. The parameters used for the RND module is given in Table S9.

Unlike prior work [21], we don't use RND with an exploration policy. Instead, we have a custom implementation using the same model-based planning backbone for RND as CEE-US, RaIR + CEE-US and RaIR. By using RND with iCEM instead of an exploration policy, we gain sample-efficiency, as also showcased in [12]. Note that although RND intrinsic reward itself is detached from the model and does not require an ensemble, we still deploy an ensemble of GNNs in this case. In previous work [12], a single GNN model was used. In order to compute the RND intrinsic reward, we take the mean predictions of the ensembles. We do this because ensembles have been shown to be more robust models [18], such that there is no inherent disadvantage to the RND baseline.

## J.2 Disagreement

The Disagreement baseline [10] uses the same intrinsic reward metric as CEE-US, given in equation (Eq. 5). We implement it with the same model-based planning backbone as in CEE-US, replacing the exploration policy in [10]. The only difference to CEE-US is that the planning horizon during free play is only 1-step. The model and iCEM controller settings are otherwise the same as for CEE-US, RaIR + CEE-US and RaIR. Essentially, the comparison between this baseline and CEE-US showcases the importance of multi-step planning for future novelty. Note that the planning horizon

is only 1 for the free play phase. For fair comparison, we use the same horizon when solving the downstream tasks as for all other methods.

# K Discussion on Goal-conditioned RL for Unsupervised Exploration

In our setup with model-based planning for unsupervised RL, the focus is on distilling knowledge into a world model that is robust enough such that we can plan for diverse downstream tasks afterwards. In the case of the goal-conditioned RL literature, the focus is on distilling this knowledge into a goal-conditioned policy, as in [25, 37, 38].

Although the exploration itself is unsupervised and guided by intrinsic rewards, the moment we solve a downstream task we need a goal specification from the environment. Whether you use a goal-conditioned policy to solve it or as in our case, use this goal to compute a reward function, is a separate distinction.

In goal-conditioned RL, you also need access to rewards at training time: the goals are used to compute this reward signal that is used to train the policy and value function. In a way, GCRL is supervised in terms of how you specify your goals (which goal space you use) and the rewards with respect to these goals. We consider the reward function in GCRL to be actually known to the agent. The nice part in GCRL is, you can put in very general supervision with minimal assumptions, that don't require much knowledge of environment dynamics. However, doing so, GCRL is restricting the class of tasks you can solve, namely to tasks that can be described as a goal. But in a wide range of environments, it is hard to specify tasks as a goal: for example running as fast as possible in Halfcheetah.

Our goal is to learn a robust world model that we can use for planning in a later extrinsic phase. In our setup, during the intrinsic phase we make similar assumptions as Plan2Explore [11]. For the extrinsic phase, we need some specification of the task. For CEE-US, a reward function is provided at test time, whereas for GCRL it is a goal (which in fact also implies a reward function).

In model-based planning, since we have complete freedom of choice for the reward function, we can specify any task that can be solved in an MDP setting. Note that we can use our setup to also optimize a goal-conditioned sparse reward. The challenge for us is the limitedness of the zero-order trajectory optimizer: solving long horizon tasks with just sparse rewards becomes challenging, especially with imperfect world models. That's why in practice, we fall back to dense rewards for planning.

# L Connections between Compression and RaIR

Our regularity objective, that seeks out low-entropy states with high redundancy, shares close ties with compression, and specifically with lossless compression using entropy coding.

We implemented a version of our regularity idea using the lossless compression algorithm `bzip2` corresponding to the direct version of RaIR with order $k = 1$ (Sec. 2.2). In this case, we describe the state $s$ directly by the properties of each of the entities and the function $\phi : \mathcal{S}_{\text{obj}} \to \{\mathcal{X}\}^+$, that maps each entity to a set of symbols and obtain $\Phi(s) = \Cup_{i=1}^{N} \phi(s_{\text{obj},i})$ as a union of all symbols for $N$ objects. Instead of computing the entropy for the frequencies of occurrence in the resulting multiset of symbols like in RaIR, we instead transform these symbols into bytes and compress them with `bzip2`. We then define the intrinsic reward for compression as the negative length of the compressed ByteString such that:

$$r_{\text{compression}} = -\,\text{len}(\texttt{bzip2.compress}(\{\Cup_{i=1}^{N} \phi(s_{\text{obj},i})\}^+\texttt{.tobytes()})). \tag{S5}$$

We also managed to create regular shapes and patterns when optimizing for $r_{\text{compression}}$ via planning with ground truth models and also for free play with learned models. The reason we chose not to pursue this direction was because 1) lossless compression algorithms like `bzip2` don't perform as well on short ByteStrings, which is the case for us, as e.g. in CONSTRUCTION, we compress only 6 objects with their corresponding $x$-$y$ positions, 2) artifacts are introducted to the regularity/compression metric by the transformation into bytes, where certain symbols become more compressible than others in this representation without any added regularity. As a result, we preferred our formulation with RaIR as it provides better control over the generated patterns and structures.

# M Discussion on Object-centric Representations

We want to further discuss our reliance on object-centric or more broadly entity-centric representations of the world for our regularity computations. We view the world as a collection of entities and their interactions. This is a very general assumption, and one that is commonly used for its potential to improve sample efficiency and generalization of learning algorithms in a wide range of domains [46–48]. The principles underlying entity segmentation and their importance for our perception are also studied in cognitive science [49–51].

Entity itself is a very abstract and fluid concept. For us as humans, it dynamically changes depending on the granularity of the control problem we want to solve. For example, if we want to move a plate of cookies from the counter to the table, each cookie on the plate is not treated as a separate entity, instead we view the whole plate as one. If we later want to eat the cookies, then the entity-view changes: now each cookie is its own entity. (Example adapted from a private discussion with Maximilian Seitzer.)

Deciding on which level of abstraction to use in which control setting is non-trivial and a research direction which we find very exciting. In object-manipulation environments that are currently encountered in RL (with hard rigid bodies and non-composite structures), we don't find much ambiguity on entity identifications, hence our statement in Sec. 6 that observations per object in these cases are naturally disentangled. However, in open-ended real world scenarios it is indeed needed to dynamically choose the right level of abstraction in the perception of a scene and identify the necessary set of entities for the control problem at hand. Using RaIR for a real-world e.g. cluttered scene would be very challenging without closing this action-perception loop. With orthogonal research in visual perception, we are hopeful that this will be possible and we see these synergies as exciting future work.

# N Code and Compute

Code is available on the project webpage `https://sites.google.com/view/rair-project`. We run the ground truth model experiments on CPUs. As we are using the true environment simulator as a model, each imagination step in the planning horizon takes as long as an environment step. We parallelize the ground truth models on 16 virtual cores The controller frequency in this case is ca. $0.25\,\text{Hz}$, for the settings given in Table S4. For the free-play phase, we have a fixed number of transitions collected at each training iteration, which get added to the replay buffer. After the data collection, the model is trained on the whole replay buffer for 25 epochs. Since the buffer size increases at each training iteration with newly collected data, for this fixed number of epochs, the corresponding number of model training updates and thus also the runtime of the iteration, increase throughout free play. For RaIR + CEE-US, the full free-play (300 training iterations) in CONSTRUCTION with 6 objects, where overall 600K data points are collected, takes roughly 87 hours using a single GPU (NVIDIA GeForce RTX 3060) and 6 cores on an AMD Ryzen 9 5900X Processor. The majority of this time is spent on the model training after data collection. The controller frequency for the collected rollouts with RaIR and the epistemic uncertainty calculations using a GNN ensemble is ca. $5\,\text{Hz}$.