# OpenReview forum: "Regularity as Intrinsic Reward for Free Play"
_NeurIPS.cc/2023/Conference — NeurIPS 2023 poster_

### Official Review · Reviewer_fHtQ · 2023-06-11

**Soundness:** 2 fair
**Presentation:** 3 good
**Contribution:** 2 fair
**Rating:** 5
**Confidence:** 5

**Summary:**

This work proposes an exploration approach that prioritizes "regularity" in the exploration behavior, in contrast to typical novelty-seeking exploration. The objective is defined as the minimization of entropy over user-defined, object-centric features. The method is primarily evaluated on block rearrangement in the robotic manipulation setting.

---
**8/22/23 Update after author response**

My major concern with this paper was the generality of RaIR. In my opinion, there are two ways to create a good exploration method. The first is to come up with a general method with little-to-no assumptions that works across a variety of tasks. The second is to create a specialized method, be upfront about the assumptions made, and get very good performance by leveraging those assumptions.

Before the rebuttal, RaIR was introduced as a general exploration approach, yet its experiments were limited to stacking objects of identical shapes and sizes. While stacking is a very hard exploration challenge, I expected to see environments beyond object manipulation like locomotion, video games, etc. Furthermore, RaIR does depend on the key assumption of object-centric state representation, which it uses in both its objective and dynamics model.

The authors provided some followup experiments in the rebuttal. They evaluated their method on a more generalized version of their construction environment, as well as on a Quadruped locomotion environment. They also provided some additional new environments (RoboDesk, Walker),  but did not run exploration experiments on them, only sanity checks.

So to conclude:

**My concern on generality of the method:** With the new exploration experiments on generalized construction and quadruped, the authors have shown that RaIR can handle diverse objects, and can work in a single non-manipulation task. Therefore my concern is somewhat alleviated. I would recommend the authors to show more diverse environments in general though, to remove any remaining doubt from future readers that RaIR is not engineered towards a particular type of task.

**My other concerns about object-centric assumption, baselines, related work, novelty, etc:** The authors have done a good job on addressing my concerns here. I hope these discussions will go into the next version of the paper, and influence how RaIR is presented.

As of such, I will raise my score to 5. The authors have passed the threshold for addressing my main concern on generality, given the limited rebuttal time.

For future improvements, I would recommend the following:
1) Continue adding more diverse environments to show RaIR is a general exploration method.
2) Study the assumption of object-centric representation a bit more - what happens if we use RaIR with an unstructured dynamics model? Will the regularity objective still work?

**Strengths:**

- The high level idea of using regularity to focus exploration is well motivated, and addresses the weaknesses of the more common novelty-seeking exploration methods.
- This method explores well in block stacking / rearrangement, a hard exploration problem. The experimental section analyzes the block stacking task extensively.
- The manuscript is generally well written.

**Weaknesses:**

My main concern is over the scope and generality of this method. While the introduction motivates regularity as a general-purpose exploration objective, the actual implementation and assumptions made to implement the objective restricts it to a very particular set of tasks - manipulation of identical and well characterized objects.  I would have expected, from the introduction, a method that could apply to all sorts of domains (e.g. robotic locomotion, video games) instead of only object manipulation tasks.

In RaIR's environments, all objects are the same shape and have the same physics. For the construction environment, the authors only apply RaIR to the x-y state space to bias towards vertical alignments. However, this x-y assumption would break if the objects were not identical, since the vertical alignment would now be object dependent.

Another major limitation (which I appreciate the authors mentioning in Sec. 5)  seems to be the assumption of a known state space and dependence on user features. RaIR seems to hinge on having a state space that is interpretable, so that the user can provide user-defined features to bias the exploration.  I am concerned about the scalability of providing user defined features, particularly if the state space becomes more complex, or uninterpretable (e.g. pretrained ImageNet representations). However, if RaIR claims to be a general exploration method, it should address such problems.

Finally, the experiment section is lacking baselines. There is only 1 novelty-seeking baseline that the authors compare against, which is by design going to be suboptimal in assembly tasks. I would like to see how RaIR compares to an exploration baseline that actually seeks regularity in its objective as well. I believe SMiRL (1) to be a viable baseline, as it also seeks to minimize surprise. I would also like to see some baselines from the unsupervised goal-conditioned RL literature, as those methods also do stacking.

## Moderate concerns:
- Claims of Novelty - The authors claim in the first line of the abstract that they propose regularity as a novel reward signal for intrinsic motivation. However, the high level idea of regularity / stability / niche-seeking has been explored as an intrinsic motivation signal for RL(1), and more generally in the active inference literature (2). I would like to authors to revise their claims of novelty, and include the relevant literature.
- Related work is rather short. Only a small paragraph is dedicated to intrinsic motivation. A rather long paragraph is dedicated to Compression, which is not necessary.  I would recommend the following changes.
	- Expand on intrinsic motivation in RL, particularly methods that also do manipulation tasks and how RaIR compares to them. Mention (1,2) as well, as they also explicitly mention concepts similar to regularity (stability, niche-seeking). Another omission is the unsupervised skill discovery literature, where some works (3) also show manipulation tasks.
	- Paragraph on compression is unnecessarily long and detailed - it is sufficient to briefly mention compression progress and move the rest to the appendix if necessary. That space should be used mainly on Intrinsic Motivation.

## Minor concerns:
- Line 84: what is $n_a, n_s$?
- Line 124: I found the "Relational RaIR of order K" to be a bit abstract, and I needed to read it a few times over to fully understand it. Having a figure like Figure 3 for this section would have been very helpful.
- Line 220: Please point to a figure.
- Recreation experiment - while the recreation behavior is interesting, the authors do not include any practical applications or downstream implications for this phenomenon. I would prefer this section be moved to the appendix and the space be used for other things (e.g. experiment in new domain) or extended related work.

1. Berseth et al, SMiRL: Surprise Minimizing Reinforcement Learning in Unstable Environments, ICLR 2021
2. Friston et al, Active inference and learning. Neuroscience & Biobehavioral Reviews 2016
3. Zhao et al,  Mutual Information State Intrinsic Control, ICLR 2021

**Questions:**

What is the scope of RaIR? I can see two ways to present RaIR.

One way, and the way the paper is written, is to promise a broad scope. However, can RaIR be applied to more complex and realistic settings both in the manipulation domain and beyond it?  I would suggest for experiments that show RaIR can be applied to other domains, and in more complicated scenarios.

The other way, is to narrow down the scope and claims of RaIR to an intrinsic reward for object manipulation, which itself is a very challenging and interesting domain. I would like to see experiments in more realistic manipulation setups (different shapes, sizes, masses, etc.), as mentioned above.

**Limitations:**

The authors did address a main limitation, which is the assumption to an object centric, interpretable state space.
Another limitation not mentioned is the need to define task-specific features for RaIR.

---

> ### Author Rebuttal · Authors · 2023-08-09
>
> We thank the reviewer for their detailed feedback on our work. We address their concerns in the following.
>
> # Generality of RaIR
>
> We have provided new experiments in new domains in the rebuttal pdf (Fig. D2-D4). As the reviewer suggested we have included RaIR generated arrangements in locomotion domains (Quadruped and Walker), Construction with custom shapes and RoboDesk.
>
> For Quadruped and Walker, we take the hips, ankles, toes etc. to be the entities over which we compute RaIR. We see the emergence of interesting regular poses, that also happen to coincide with goal poses proposed in the Roboyoga benchmark [1].
>
> For Custom Construction with shapes, we include not just cubes (mass(m) = 2), but also a flat block (m=1.5), a short column (m=1) and a ball (m=1). In this case, we compute RaIR again on the objects’ x-y positions (center of mass (CoM)). We show that we are still able to generate stacks and regular constellations. We would also argue that there is no limitation within RaIR that would cause it to not work for objects with different shapes. If we have very complicated geometries, e.g. composite shapes, we might want to represent each entity not just with their CoM but instead add new keypoints.
> We also showcase RaIR in Robodesk where the entities are very varied: drawer, slide, buttons and two different blocks and a ball. Here, with regularity, we observe the agent using the ball or the block to press buttons and stacking objects together. We hope that the new experiments better illustrate the generality of RaIR as a concept.
>
> *RaIR seems to hinge on having a state space that is interpretable*
>
> We refer the reviewer to our general response, where we discuss this at length. We want to reiterate here shortly that we use proprioceptive information for objects (positions, orientation and velocities), which itself is not user-defined and is specified by the environment. We only consider access to the knowledge which subspace corresponds to object positions. We find the inference of object positions from images to be an orthogonal research problem with ongoing works showing promising results. Note that using ImageNet features alone does not help solve the control problem, as the focus there is on semantic understanding for visual learning tasks [2]. In this case, additions are needed to align the representation and actions, i.e. control over the environment. One could create regularity spatially with the added constraint of compressing ImageNet features (e.g. object labels), similar to the color experiment we show in Fig D1.
>
> # Novelty of RaIR
>
> There is a fundamental difference between our notion of regularity and niche-seeking proposed in SMiRL and active inference literature: in all the environments showcased in SMiRL the environment itself is dynamic (or unstable) and the agent wants to reach stable and familiar states. For example, not fall down the cliff, avoid enemies that are shooting at you. Our notion of regularity is completely decoupled from surprise and is in fact closely related to compression: we want to find a state description of the current scene that is regular and compressible within itself.
>
> Methods such as SMiRL require an active source of entropy. In an environment where the environment itself is static unless you act on it, which is the case in all environments we consider, we strongly believe SMiRL wouldn’t work due to the Dark Room problem, and the agent would avoid doing anything. This is also what happens when you penalize novelty in object-manipulation environments: the agent doesn’t touch any objects to make sure everything is predictable.
>
> We will revise the related work section to make these distinctions clearer as suggested by the reviewer, and extend the related work discussion to include skill literature as well.
>
> # Baselines
>
> We have included new baselines in the rebuttal pdf, most notably RND (see general response for more details). RND aims for state space coverage and can be seen as an extension of count-based metrics to continuous domains, thus has no preference for chaos. We showcase that it performs worse than both CEE-US and our regularity augmented version RaIR + CEE-US on the Singletower 3 task.
>
> It is important to highlight here that the main goal of our work is to introduce RaIR as an additional reward signal to drive exploration towards regularities. As such, RaIR in principle can be used to augment different intrinsic reward functions. In our paper, we illustrated this on the example of CEE-US+RaIR as CEE-US was shown to beat the other baselines (both policy and planning based) in [2].
> Note that in unsupervised goal-conditioned RL paradigms, the driving exploration force is still commonly ensemble disagreement, as in LEXA[1] and PEG[4]. In Sec. 4, we did mention PEG and its stacking performance in a similar environment. In principle, RaIR could also be plugged into these methods as a goal-picking strategy, augmenting ensemble disagreement.
>
> # Addressing minor concerns
> - $n_a$ is the action space size and $n_s$ is the state space size.
> - Abstract order-k: We indeed aimed to generalize RaIR and abstract away from the mapping functions such as distance or difference (with k=2) to functions that operate over k-tuples corresponding to k objects. In Figure 2b, you can find an illustration for k = 2 that we think is useful to clarify k>2.
> - Line 220: We will point to Figure 1 as the reviewer suggested.
> - Recreation experiment: Our goal with these experiments was to showcase that we have a way to automatically pick-up regularities in the environment as you observe them.
>
> [1] Mendonca et al (2021), Discovering and Achieving Goals via World Models. NeurIPS
>
> [2] Sharma et al (2023),  Lossless Adaptation of Pretrained Vision Models for Robotic Manipulation. ICLR
>
> [3] Sancaktar, et al (2022). Curious exploration via structured world models yields zero-shot object manipulation. NeurIPS
>
> [4] Hu, et al (2023). Planning Goals for Exploration.

---

> > ### Comment · Reviewer_fHtQ · 2023-08-11
> > **Appreciate the progress; Still doubts remaining.**
> >
> > I thank the reviewers for their efforts so far, it is a good start.
> >
> > On my concern of generality - the new environments with increased variation and results of RAIR with Ground Truth dynamics models is promising. To be fully convincing, the authors should follow the same experiment protocol as the previous experiments, e.g. learn the model while training, and have some sets of evaluation metrics instead of qualitative snapshots. A "nail-in-the-coffin" experiment would be to use real objects meshes, e.g. from the YCB dataset, and show RAIR can find interesting configurations.
> >
> > On my concern about novelty with respect to entropy-minimization methods like SMIRL - I still believe RAIR and SMIRL are similar due to their entropy minimization objective for exploration. Just like SMIRL, I would not expect RAIR to work in static environments without some external force or exploration policy driving disturbances (for RAIR, you are using P2E to do this).
> >
> > Is it fair to say that if we ran SMIRL with a novelty seeking bonus, this method will be similar to RAIR without a user defined prior over what dimensions to compute entropy over? If so, then I think it will be natural to run a "SMIRL-like" baseline that is RAIR with entropy minimization over all dimensions.
> >
> > On my concern about how user defined state-spaces are not general - the authors claim that the object information (e.g. object orientation) is defined by the environment and not the user. However, the environment can arbitrarily decide the object orientation which can be a problem.
> >
> > Let us consider a cylinder in the stacking case. There are two possible ways to align the XYZ frame - align the XYZ axis so the up vector points out of the flat circle of the cylinder, or align the XYZ frame so the up vector points out of the rounded edges.  (I attempted to add 2 drawings for illustration).
> >
> > ```
> > 0===0 -->
> > ```
> >
> > ```
> > ||
> > --->
> > ||
> > ```
> > So if this frame is arbitrarily decided by the environment, then it will be hard, if not impossible, for the user to define something like a "XY" prior since the frames are different per object.
> >
> > Finally, a minor comment - "proprioceptive" information means the robot body info (e.g. joint positions, velocity) and not info about external objects.

---

> > > ### Author Response · Authors · 2023-08-13
> > >
> > > We thank the reviewer for their quick response. In this response we would like to clarify a few things.
> > >
> > > ### Comparison to SmiRL
> > >
> > > The distributions over which we compute entropy compared to SmiRL are completely different. In SmiRL, the agent regards the entropy of its state marginal distribution under its current policy $\pi_{\phi}$ at each time step. This is done by fitting a density model with parameters $\theta$ over the environment steps collected in that rollout / episode ($\{s_0, ….s_{t-1}\}$) so far and the reward is computed as the log probability $\log p_{\theta_{t-1}}(s_t)$. In summary, SmiRL seeks **familiar** states that are seen in the data collected so far  **within that rollout**. We tried SmiRL reward with ground truth models in our manipulation environment and in the quadruped(ant) locomotion environment. In the manipulation environment it simply lets everything stay as it is and does nothing. In the quadruped environment, it tries to go back to the initial state or the first stable pose that is reached. There is no incentive to be stable in a regular or symmetric pose. Note that the authors of SmiRL themselves promote this reward signal for **unstable** environments.
> > >
> > > We also want to clarify a misunderstanding: **RaIR without any novelty-seeking objective still works**. For free play, we showcase results with pure RaIR ($\lambda=0$) in Figure D6 in the rebuttal pdf as well as the supplementary figures S6, S7 and Table S1. RaIR alone can solve the stacking 3 objects task with 64% success rate, better than pure ensemble disagreement. Performance improves when we add ensemble disagreement in RaIR+CEE-US, because 1) disagreement helps exploration via the information gain objective 2) since the agent targets the places where it disagrees, i.e. has high epistemic uncertainty, it becomes its own adversary and this in our experience leads to a more capable world model (which can also be seen in comparison to the RND baseline using the same backbone).
> > >
> > > In the setting with ground truth models, all configurations are obtained by only optimizing for RaIR. SmiRL would not be able to be used in this way.
> > >
> > > SMiRL with novelty seeking bonus is NOT similar to RaIR. SmiRL just makes sure a certain configuration is not altered or reoccurs during a rollout, so a constancy bias.
> > >
> > > And we want to emphasize again that in our work regularity refers to the repetition of a certain pattern, such that there is redundancy in the description of a state. One type of redundancy is symmetries. Even an unstable configuration, e.g. a vertical headstand in the air in Walker environment, has high RaIR because it is a symmetric state, even though the agent won’t be able to keep the pose.
> > >
> > >
> > > ## Concern about user-defined state spaces
> > >
> > > We believe there is a misunderstanding: First of all, we would like to mention that for regularity as such, there is no universal representation or universal measure. It is similar to the *no free lunch* principle --  you have to commit to a particular inductive bias. Whether it is the environment or the agent deciding on this, is more of a philosophical question.
> > > Nevertheless, if object positions are given in a certain frame of reference, we will either find regularity in this frame (world coordinate frame or agent centric one) when direct RaIR is used,  and with relational RaIR (default case in our work) we are independent of the frame (as long as all objects are represented in the same frame of reference). Our symbols are the difference vectors between the center of masses of different entities.
> > >
> > > Currently, we are not considering orientation as a symbol. Note that even when RaIR is orientation-agnostic, finding a stable configuration works because if you put the cylinder in a configuration where it doesn’t roll over, such that you can stack another object on top, you have much higher RaIR.
> > >
> > > In the example you gave, when considering orientations, then simply a change in orientation of  the lower cube would yield a higher RaIR, so this could be found by the agent.  If the representation of orientation is adversarial, such that alignments would yield a hard-to-stack configuration, then orientation can be added in as a secondary constraint in hierarchical RaIR: compute RaIR once for only positions and once for `concat([positions, orientations])`, and sum these two costs. In this case regularity in orientation is attempted in addition, if possible.

---

> > > > ### Comment · Reviewer_fHtQ · 2023-08-14
> > > >
> > > > Thank you for the response.
> > > >
> > > > I acknowledge that I may have some misunderstanding about RaIR. I see how SMIRL differs from RaIR now, but what is still unclear to me is why would RaIR work in the block stacking case without any novelty-seeking reward? Could the author explain how RaIR without some extrinsic exploration force would discover block stacking more in detail? In particular, wouldn't RaIR also enforce a constancy bias, just over particular dimensions of the state space?
> > > >
> > > > Finally, the discussion on the user-defined state space the authors provided clears up my concern about it.

---

> > > > > ### Author Response · Authors · 2023-08-15
> > > > >
> > > > > We thank the reviewer for their response and we are glad that we were able to clarify their questions. We are happy to further elaborate on RaIR without novelty-seeking rewards.
> > > > >
> > > > > We would call RaIR a redundancy bias and not a constancy bias in the state description. A constancy bias implies that RaIR tries to keep the relations between objects constant, whereas the goal is to make these relations repeat, i.e. if objects 1-2 and 1-3 are in the same relative position to each other, there is a redundancy, which is what RaIR maximizes for. The goal is to get to the state with the highest redundancy.
> > > > >
> > > > > That’s why with RaIR, there is an incentive to try to build towers and other regular patterns starting from a randomly reset environment state. This is what we showcase in the generated patterns, when we plan for only high RaIR with Ground Truth models in Figure 1 in the main paper. Note that RaIR has a step-wise landscape: when a new object is aligned, i.e. becomes part of a pattern, we get a step-wise increment in RaIR (see figure 4 in the main paper). These improvements are the signals that the zero-order trajectory optimizer picks up.
> > > > >
> > > > > In the same way, even when we only use RaIR as an intrinsic reward in free play, the learned models also attempt to create regular structures, such as stacks. As a result, these types of dynamics, e.g. objects on top of each other, are better explored throughout free play with RaIR, amounting to better downstream task performance in assembly tasks compared to pure ensemble disagreement.
> > > > >
> > > > > In free play, we start out with randomly initialized world models. One big challenge is: we first need to learn object dynamics to be able to manipulate them and plan for e.g. regular states with RaIR. In this case, explicit novelty-seeking objectives help accelerate learning. For pure RaIR (without the exploration force of CEE-US), the learning of object dynamics is slower in the beginning. However, once these dynamics predictions improve, RaIR alone can yield plans for regular states such as stacks. That’s why RaIR downstream task performance starts picking up later into the free play as shown in Figure S6, but then consistently outperforms pure disagreement in assembly tasks.
> > > > >
> > > > > In summary, pure RaIR as an intrinsic reward is still a powerful and useful signal for free play as reflected in the downstream task performance. And with the injection of ensemble disagreement in RaIR+CEE-US, it becomes even more powerful, accelerating learning in free play.
> > > > >
> > > > > We hope this explanation clears up the questions of the reviewer. We are happy to answer further questions.

---

> > > > > > ### Comment · Reviewer_fHtQ · 2023-08-16
> > > > > > **Empirical confusions cleared up, remaining concerns over assumptions of test time reward and object-centric tasks.**
> > > > > >
> > > > > > Thank you for the clarification. My concerns over the implementation and empirical results of RaIR are cleared up, except for the full results on the more general environment, which I hope the authors are working on. Raised my score to a 4, and I will reconsider the score in light of full experimental results.
> > > > > >
> > > > > > My remaining concerns are more philosophical, and are as follows.
> > > > > >
> > > > > > ### Assumption of access to a test time dense reward.
> > > > > > I would also like to point out another limitation / assumption that I didn't quite notice earlier. This work focuses on unsupervised exploration for building a robust world model, and then assumes access to the reward function of the target task for planning.
> > > > > >
> > > > > > Many unsupervised exploration works that I am familiar with, particularly the GCRL exploration literature, do not assume access to the test time reward function, and distill a goal-conditioned policy that only require a test goal state to achieve. Therefore, these methods make less assumptions about the task at test time.
> > > > > >
> > > > > > This limitation isn't strictly worse, it can be a feature - if the user has access to the task reward function, RaIR can exploit this. However, this falls short of the fully unsupervised RL case, where the RL agent not only needs to explore, but also distill useful skills without knowledge of the test tasks.
> > > > > >
> > > > > > ### Are object centric representations really general?
> > > > > >
> > > > > > > We embrace object-centric representations as a suitable inductive bias in RL, where the observations per object (consisting of poses and velocities) are naturally disentangled.
> > > > > >
> > > > > > I would say that object-centric representations limit the space of tasks to environments with rigid bodies and task definitions where a physical object is equivalent to a semantic object in the task definition. So tasks like "pick up apple" where apples are clearly the object work, but tasks like "pick up container of sand" may break down because it is hard to build an object detector for loose sand.  So while a large space of manipulation tasks can be cleanly represented with object-centric representations, it is clear that object-centric representations are not suitable general representation for all tasks, particularly those where the notion of "object" is not defined well.
> > > > > >
> > > > > > Therefore, I still view RaIR, which depends on object-centric features, and its structured world model, as still somewhat limited in generality because of the object-centric representation.
> > > > > >
> > > > > > For addressing these more philosophical concerns, I look forward to some constructive discussion about this, e.g. pros and cons of a side, over attempts to convince me that one way or another is the "right" way. I also hope some of this discussion will make it into the paper.

---

> > > > > > > ### Author Response · Authors · 2023-08-19
> > > > > > > **Response to Reviewer fHtQ [1/2]**
> > > > > > >
> > > > > > > We thank the reviewer for increasing our score.
> > > > > > >
> > > > > > > ### Free Play Experiments in General Environment
> > > > > > >
> > > > > > > We have run free play experiments in the new Construction environment with diverse shapes: one cube, one flat block, one short column and one ball. We show the zero-shot performance gains we get with RaIR augmentation (RaIR+CEE-US) vs. pure ensemble disagreement (CEE-US) in the following table (evaluated at free play iteration 200):
> > > > > > >
> > > > > > > | Task@200    | Tower 2: Cube (bottom) - Ball (top) | Tower 2: Cube(bottom) - Flat Block(top) | Tower 3: Cube(bottom) - Column - Ball(top) |
> > > > > > > |-------------|-------------------------------------|-----------------------------------------|--------------------------------------------|
> > > > > > > | RaIR+CEE-US | **0.57 ± 0.04**                     | **0.63 ± 0.01**                         | **0.12 ± 0.02**                            |
> > > > > > > | CEE-US      | 0.22 ± 0.07                         | 0.38 ± 0.01                             | 0.02 ± 0.01                                |
> > > > > > >
> > > > > > > Note that these tasks are very challenging as stacking in this environment requires balancing these objects which have different dynamics. Here in this general environment, we again see the benefit of regularity as an intrinsic reward signal. We hope these new experiments further showcase the generality of our proposed method.
> > > > > > >
> > > > > > > Next, we would like to discuss the additional topics raised by the reviewer.
> > > > > > >
> > > > > > > ### Assumption of access to a test time dense reward
> > > > > > >
> > > > > > > It is indeed two different approaches. In our case, the focus is on distilling knowledge into a world model that is robust enough such that we can plan for diverse downstream tasks afterwards. In the case of the GCRL literature, the focus is on distilling this knowledge into a goal-conditioned policy.
> > > > > > >
> > > > > > > Here we want to make some clarifications:
> > > > > > >
> > > > > > > Although the exploration itself is unsupervised and guided by intrinsic rewards, the moment we solve a downstream task we need a goal specification from the environment. Whether you use a goal-conditioned policy to solve it or as in our case, use this goal to compute a reward function, is a separate distinction.
> > > > > > >
> > > > > > > In goal-conditioned RL, you also need access to rewards at training time: the goals are used to compute this reward signal that is used to train the policy and value function. In a way, GCRL is supervised in terms of how you specify your goals (which goal space you use) and the rewards with respect to these goals. We consider the reward function in GCRL to be actually known to the agent. The nice part in GCRL is, you can put in very general supervision with minimal assumptions, that don't require much knowledge of environment dynamics. However, doing so, GCRL is restricting the class of tasks you can solve, namely to tasks that can be described as a goal. But in a wide range of environments, it is hard to specify tasks as a goal: for example running as fast as possible in Halfcheetah.
> > > > > > >
> > > > > > > Our goal is to learn a robust world model that we can use for planning in a later extrinsic phase.
> > > > > > > In our setup, during the intrinsic phase we make similar assumptions as Plan2Explore. For the extrinsic phase, we need some specification of the task. For CEE-US, a reward function is provided at test time, whereas for GCRL it is a goal (which in fact also implies a reward function.)
> > > > > > >
> > > > > > > In model-based planning, since we have complete freedom of choice for the reward function, we can specify any task that can be solved in an MDP setting. Note that we can use our setup to also optimize a goal-conditioned sparse reward. The challenge for us is the limitedness of the zero-order trajectory optimizer: solving long horizon tasks with just sparse rewards becomes challenging, especially with imperfect world models. That’s why in practice, we fall back to dense rewards for planning.
> > > > > > >
> > > > > > > Hope this discussion is helpful to clarify the pros and cons for these two paradigms. We are also very happy to include the essence of this discussion in the related work section that we have extended as per the reviewer’s suggestion.
> > > > > > >
> > > > > > > We discuss the second point raised by the reviewer in the second part of our answer below.

---

> > > > > > > > ### Author Response · Authors · 2023-08-19
> > > > > > > > **Response to Reviewer fHtQ [2/2]**
> > > > > > > >
> > > > > > > > ### Are object centric representations really general?
> > > > > > > >
> > > > > > > > We find this topic of discussion very exciting. Perhaps it is best to abstract away from objects and say that we view the world as a collection of entities and their interactions. This is a very general assumption, and one that is commonly used for its potential to improve sample efficiency and generalization of learning algorithms in a wide range of domains [1-3]. The principles underlying entity segmentation and their importance for our perception are also studied in cognitive science [4-6].
> > > > > > > >
> > > > > > > > *Entity* itself is a very abstract and fluid concept. For us as humans, it dynamically changes depending on the granularity of the control problem we want to solve. E.g., if I want to move a plate full of cookies from the counter to the table, I don’t treat each cookie as a separate entity, but view the plate with all cookies as one. If I want to later eat the cookies, then my entity-view changes: now each cookie is its own entity. Similarly, in the sand container example, unless we want to use the sand particles to spray a pattern etc. we don’t need to model each particle individually.
> > > > > > > >
> > > > > > > > Deciding on which level of abstraction to use in which control setting is non-trivial and a research direction which we find very exciting. In object-manipulation environments that are currently encountered in RL (with hard rigid bodies and non-composite structures), we don’t find much ambiguity on entity identifications, as the reviewer also alluded to, hence our statement that observations per object in these cases are naturally disentangled.
> > > > > > > >
> > > > > > > > However, in open-ended real world scenarios it is indeed needed to dynamically choose the right level of abstraction in the perception of a scene and identify the necessary set of entities for the control problem at hand.
> > > > > > > >
> > > > > > > > Using RaIR for a real-world e.g. cluttered scene would indeed be very challenging without closing this action-perception loop. With orthogonal research in visual perception, we are hopeful that this will be possible and we see these synergies as exciting future work. We will also include this discussion in the limitations section of our method in the revised paper.
> > > > > > > >
> > > > > > > >
> > > > > > > > [1] Locatello et al (2020), Object-Centric Learning with Slot Attention, NeurIPS.
> > > > > > > >
> > > > > > > > [2] Battaglia et al (2016), Interaction networks for learning about objects, relations and physics, NeurIPS.
> > > > > > > >
> > > > > > > > [3] Tsividis et al (2021), Human-Level Reinforcement Learning through Theory-Based Modeling, Exploration and Planning (Preprint).
> > > > > > > >
> > > > > > > > [4] Spelke (1990), Principles of Object Perception, Cognitive Science.
> > > > > > > >
> > > > > > > > [5] Spelke et al (1993), Gestalt relations and object perception: A developmental study, Perception.
> > > > > > > >
> > > > > > > > [6] Peters & Kriegeskorte (2021), Capturing the objects of vision with neural networks, Nature Human Behavior.

---

> > > > > > > > > ### Comment · Reviewer_fHtQ · 2023-08-19
> > > > > > > > > **Thank you for the discussion. Would still like to see some more results.**
> > > > > > > > >
> > > > > > > > > I thank the reviewers for the responses to my concerns about the dense reward at test time and object-centric representations.
> > > > > > > > >
> > > > > > > > > To add onto the discussion about test time rewards, I will say that many GCRL works like LEXA and its descendants use self-supervised rewards that do not require reward engineering. These methods are able to perform locomotion and manipulation skills like stacking, although perhaps not to the level of RaIR with its dense reward for stacking. It is not hard to specify running skills with goals, because the goal can be a target velocity, or a demonstration, or even language - just that empirically, most GCRL works choose environment positional state as the goal representation. See [1] for a GCRL method that performs numerous locomotion tasks.
> > > > > > > > >
> > > > > > > > > I like the discussion on object-centric representation, indeed choosing the right level of task abstraction is crucial for object-centric methods to work well. Whether using this strategy with object-centric representations, or using a non-object-centric representation (e.g. completely latent representation trained from big data), for solving general tasks, is an open research question.
> > > > > > > > >
> > > > > > > > > ## Remaining issue: incomplete results on the new envs.
> > > > > > > > > However, perhaps the most pressing issue for me still, is the unfinished general experimental results. At the beginning of rebuttal, to showcase the generality of RaIR, the authors have proposed several additional environments (Walker, Quadruped, RoboDesk) and have improved their original environments. They have shown that with a **ground truth dynamics model**, optimizing RaIR can result in desirable configurations. However, this is a sanity check rather than a real experiment - the point of RaIR is to enhance exploration data for training a dynamics model. Therefore, I expect to see experiments that train the dynamics model with RaIR exploration.
> > > > > > > > >
> > > > > > > > > The authors have provided the exploration experiment for the improved Construction environment, which I appreciate. If possible, could the authors include videos, curves, and some more analysis of the experiment?
> > > > > > > > >
> > > > > > > > > As of now, I would still like to see some more results in a similar vein on the new environments. I understand given the time and compute limitations, it may not be possible to run full experiments on all tasks. However, I hope the authors will try their best to showcase a subset or intermediate results on more tasks to better showcase the generality of RaIR as an exploration method.
> > > > > > > > >
> > > > > > > > > [1] Hafner et al (2022), Deep Hierarchical Planning from Pixels, Neurips.

---

> > > > > > > > > > ### Author Response · Authors · 2023-08-21
> > > > > > > > > >
> > > > > > > > > > We thank the reviewer for their response.
> > > > > > > > > >
> > > > > > > > > > As requested, we provide additional analysis on our update project webpage (https://sites.google.com/view/rair-project, the URL is the same anonymized link we had in the submitted paper, headers are purple for the new parts).
> > > > > > > > > >
> > > > > > > > > > This material includes: videos from free play in the generalized construction environment, RaIR curves throughout free play, downstream task curves for models checkpointed throughout free play as well as some videos of RaIR+CEE-US solving the downstream tasks.
> > > > > > > > > >
> > > > > > > > > > We also applied RaIR to the Quadruped environment. We ran free play experiments, and then evaluated success for the Roboyoga poses as shown below:
> > > > > > > > > >
> > > > > > > > > > | Task@175	| Balance Back | Balance Front | Attack      | Stand and Point | Stand Rotated |
> > > > > > > > > > |-------------|--------------|---------------|-------------|--------------|---------------|
> > > > > > > > > > | RaIR+CEE-US | 0.70 ± 0.02  | 0.79 ± 0.04   | 0.81 ± 0.05 | 0.85 ± 0.01  | 0.86 ± 0.0	|
> > > > > > > > > > | CEE-US      | 0.59 ± 0.05  | 0.67 ± 0.07   | 0.74 ± 0.04 | 0.80 ± 0.04  | 0.77 ± 0.05   |
> > > > > > > > > >
> > > > > > > > > > In these preliminary results, we show that we achieve better performance with RaIR+CEE-US on yoga poses in the Quadruped environment.
> > > > > > > > > >
> > > > > > > > > > Additional material is also on the updated webpage: videos from free play, RaIR curves throughout free play, for some poses downstream task curves for models and videos of RaIR+CEE-US solving the tasks.
> > > > > > > > > >
> > > > > > > > > > Note that as the free play videos and the RaIR curves showcase, we observe much more regular poses with RaIR+CEE-US during free play. In our current paradigm with model-based planning, due to the generalizability of learning a robust world model (which ties in nicely with our previous discussion), we achieve good results with also just CEE-US and get a performance boost with RaIR+CEE-US. We expect methods that rely on extracting task policies offline from the generated play data [1,2] as well as methods such as LEXA, which train goal-conditioned policies at play time with goals sampled from play data, to benefit even more significantly by using RaIR.
> > > > > > > > > >
> > > > > > > > > > We expect similar results also in the Walker environment. As the environment has very easy dynamics, we expect model-based approaches to be very strong there altogether.
> > > > > > > > > >
> > > > > > > > > > We hope these additional experiments and material we provide help clarify the remaining concerns of the reviewer.
> > > > > > > > > >
> > > > > > > > > > We want to make some additional comments:
> > > > > > > > > >
> > > > > > > > > > Our experiments with GT models serve as a proof-of-concept. Our main contribution is RaIR itself: we propose regularity as intrinsic reward and formalize this concept. We find this within itself a very important and novel contribution. We can achieve symmetric and regular structures in diverse environments by optimizing for this objective.
> > > > > > > > > >
> > > > > > > > > > In the experiments where we plan with GT models, our goal is to showcase the optima of RaIR without any model imperfections. When we then inject RaIR in free play, we bias exploration towards these regularities. The plots where we showcase the highest achieved RaIR values throughout free play exactly showcase this: the higher the RaIR value, the more we are observing the optima that we obtain with GT models during free play. This in turn means that dynamics in those subspaces are explored more with RaIR during the intrinsic phase. As a result, we achieve better downstream task performance where these types of dynamics pop up. And note that one major point in our work is that there is a substantial amount of tasks in diverse domains where regularities pop up (we believe that the new environments we added during the rebuttal, albeit some only with GT models, also better support this). The fact that we as humans are generally drawn towards regularities and symmetries, can both be seen as a cause and as an effect of this phenomenon. We showcase that current intrinsic rewards overlook these regularities, which is what we mitigate in this work by proposing RaIR.
> > > > > > > > > >
> > > > > > > > > > Overall, we showcase this trend that when we learn models on-the-go and inject RaIR, we are biasing exploration towards the RaIR optima showcased with GT models and getting better performance for tasks where regularities are of importance. There is in principle no reason for this chain of effect to not apply to the other environments we showcased. For some environments like Walker it may be too easy since dynamics are very easy to learn, or for Robodesk which requires very long-horizon planning, it may be more challenging and require more free play time, but the underlying principle remains. We hope that the experiments we managed to run in the rebuttal period make the generality of this effect clearer.
> > > > > > > > > >
> > > > > > > > > > And we also want to thank the reviewer again for their efforts and their active engagement during this rebuttal.
> > > > > > > > > >
> > > > > > > > > > [1] Lambert et al (2022), The challenges of exploration for offline reinforcement learning, Preprint
> > > > > > > > > >
> > > > > > > > > > [2] Yarats et al (2022), Don't Change the Algorithm, Change the Data: Exploratory Data for Offline Reinforcement Learning, Preprint

---

### Official Review · Reviewer_fSr3 · 2023-07-05

**Soundness:** 3 good
**Presentation:** 3 good
**Contribution:** 3 good
**Rating:** 6
**Confidence:** 4

**Summary:**

This paper presents a new approach that utilizes a regularity as an intrinsic reward, which aims to regularize, or constrain, the search space of exploration towards state regions where the entities are more likely to have structured patterns. To this end, the paper first formulates how to define regularity as an intrinsic reward and investigates several candidates that can be used as a regularity. And this intrinsic reward is combined with the disagreement-based intrinsic rewards to guide the exploration. Experiments show that optimizing this intrinsic reward (by using the ground-truth models) indeed leads to behaviors that induces structured patterns in the environments, and further show that this can be useful in case everything is learned end-to-end.

**Strengths:**

- Interesting formulation based on a good intuition. I enjoyed reading the paper.
- Clear writing helpful for understanding the method. This should be highlighted as understanding the concept would have been difficult without the clear writing.
- Experiments are conducted to support the claims by using ground-truth models and learned models. It starts from showing the potential benefit of using the proposed reward by using the ground-truth model, then shows that it can be useful with learned models, and finally shows that it can be useful for improving the performance on downstream tasks.

**Weaknesses:**

- In contrast to clear writing until the Method section, Experiments section is a bit difficult to parse, especially it is difficult to understand the metrics used in Section 3.2. Even though they are explained in the text, further emphasis and clear description on the meaning of metrics (e.g., what relative time means, how to interpret objects in the air, flipped) could be helpful.
- As the authors already highlighted in the Limitation section, this method severely depends on the accessiblity of ground-truth internal states of all the entities in the environment, which is not feasible in practice. But this is understandable as learning such information is not the main focus of the paper.
- Another limitation of this paper is the difficulty of balancing (i) the chaotic exploration from novelty-seeking intrinsic rewards. (in this paper, disagreement-based reward) and (ii) regularity-seeking intrinsic rewards. Of course there could be cases when injecting our prior of prefereing the structuredness can be helpful, but sometimes it's not, as can be seen in Throw and Flip experiments in Table 2. Would there be a way to automatically balance these two rewards, or in general, how can we control the magnitude of injecting our prior to the exploration process?
- Experiments are a bit limited as only specific type of structuredness is investigated (for simplicity).

**Questions:**

I don't have further question other than the ones described in the Weaknesses.

**Limitations:**

Yes.

---

> ### Author Rebuttal · Authors · 2023-08-09
>
> We thank the reviewer for their assessment and feedback.
>
> # Generalizability of RaIR
>
> We have now added new experiments where we showcase RaIR in different scenarios. In order to address the concern of focusing on one specific type of structuredness, we have tested a variant of ShapeGridWorld, where we added color as part of the object representation to be compressed for regularity. In this case, we not only have x-y positions of the circles as symbols to be compressed, but also have a color encoding (as a one-hot encoding), such that RaIR aims to generate regularities not just spatially but also in the arrangement of colors. Examples can be seen in the uploaded rebuttal pdf with 2 and 3 colors (Fig. D1).
>
> We have also optimized for RaIR in the DeepMind Control Suite environments: Quadruped and Walker, where we were able to obtain some interesting regular poses via optimizing for RaIR with ground truth models. Note that these poses mostly coincide with the goals in the Roboyoga benchmark [1]. This adds to our point that seeking regularity is reflected in human designers’ idea of interesting goals and is a very fundamental bias that pops up in various domains.
>
> # Balancing Regularity and Chaos
>
> As is the case with any injection of bias, we enter a trade-off where we have some advantages and some disadvantages.
>
> As the reviewer also hinted at, if we trust the fact that we want structured behavior in, we accept to sacrifice some more “chaos”-preferring behavior. This can in fact be seen as a resource-allocation problem. If the agent only has limited play time, what should it focus on to best prepare for future tasks? If we have have unlimited resources we can train for these two rewards in e.g. an alternating fashion.
>
> However, in our case, these two forces complement each other as discussed in Sec 3.2 in the main paper. Ensemble disagreement comes with the added problem of chaos, but it also helps the agent focus on more interesting and challenging patterns during free play, instead of generating predictable and boring patterns and helps finding high RaIR patterns that are sparse solutions.
>
> That is why in this work, we investigated a linear combination of these two reward terms, with a hyperparameter $\lambda$. In the rebuttal pdf in Figure D6, we have included the downstream task performance for the chosen lambda value for RaIR + CEE-US ($\lambda=0.1$), RaIR + CEE-US with smaller lambda ($\lambda=0.01$), pure RaIR ($\lambda=0$) and pure CEE-US. We can see that the behavior towards regularity vs. chaos can be controlled by this hyperparameter.
>
> Tuning this hyperparameter is not critical and a simple heuristics can be employed that brings the ensemble disagreement and the RaIR improvements values on a similar scale. The RaIR improvement steps can also be computed analytically, e.g. by considering the full entropic state to the one with two 2 entries in alignment.
>
> As ensemble disagreement goes down with more training throughout free play, one could explore options to schedule $\lambda$ accordingly. We find studying different combination strategies to be an exciting direction and leave it to future work.
>
> Regarding the questions about the experiment descriptions and a more detailed discussion on concerns about ground truth internal states, we refer the reviewer to the general response.
>
> [1] Mendonca et al (2021), Discovering and Achieving Goals via World Models. NeurIPS

---

> > ### Comment · Reviewer_fSr3 · 2023-08-18
> >
> > Thank you for your response. I don't have follow-up immediate concerns or questions. Currently I would like to maintain my score but I might potentially adjust my score after the internal reviewer discussion or after having a look at other reviews and discussions after they are all finalized (especially the discussion with Reviewer fHtQ).

---

### Official Review · Reviewer_GveK · 2023-07-05

**Soundness:** 3 good
**Presentation:** 3 good
**Contribution:** 3 good
**Rating:** 6
**Confidence:** 3

**Summary:**

The paper draws inspiration from a child’s development cycle and proposes the use of regularity as an intrinsic reward to help guide exploration in RL. It shows that existing metrics of state novelty can be combined with the proposed regularity metric to enable construction of regular structure during free play in a sample efficient manner. The proposed method is validated on experiments from two different tasks.

**Strengths:**

- The paper proposes an interesting and novel formulation of using regularity, in the form of symmetry, as an intrinsic reward to guide exploration during RL.
- The paper builds on an existing novelty-based approach that uses the epistemic uncertainty obtained from an ensemble of models to guide exploration. The authors show that RaIR can be combined with such novelty-based rewards to further improve exploration in tasks that demand regularity. However, the authors acknowledge that for tasks that are chaotic like throwing and flipping, imposing such regularity hurts the performance (Table 2).
- The paper shows impressive results for stacking towers in various configurations solely from intrinsic rewards while prior work achieves this using carefully constructed reward functions.
- The paper demonstrates the effectiveness of pretraining with the regularity-based intrinsic reward in performing zero-shot generalization to downstream tasks (Sec. 3.3).
- The authors also show a way of prompting (Sec. 3.4)  the agent to build certain structures by incentivizing it to raise regularity by building the same structure as the prompt.


**Weaknesses:**

- The authors organize the state in the form of graph nodes with each node representing a specific feature of an object or the agent. I had a few questions about this -
  - How did the authors decide on this specific state representation? Does it have any advantages that a particularly attractive?
  - It would be great if the authors could clarify how the graph is built (is it a fully-connected graph with all nodes)?
   - The current form of state representation results in the authors using GNNs. However, the same representation can be dealt with using transformers. It would be interesting to add a transformer variant of RaIR.
  - This state representation might make it difficult to apply the method to the real world where privileged state information is not available. I would be curious to hear the author’s thoughts on this. The limitations section mentions that the same operation could be done on a latent space but it's unclear if the RaIR would be applicable to such a latent space (which might not be as good as the privileged information) based state representation.
- Since the method uses iCEM for policy learning, it seems like such a method of optimizing the policy for each new configuration won’t scale with reinforcement learning. It would be great if the authors could comment on this.
- How many training steps does it take to train the transition model? This would throw some light on the sample-efficiency of the method.
What is “relative time” in Figure 5?
- In Sec. 3.4, why does the agent not arrange everything in a single line to maximize regularity? Is it because of the finite horizon optimization (meaning that tasks that need more time steps would be preferred less as compared to tasks giving higher immediate rewards)?


**Questions:**

It would be great if the authors could address the comments mentioned in the “Weaknesses” section.

**Limitations:**

The paper does have a limitations section. The authors must also acknowledge the following from the “Weaknesses” section -

```This state representation might make it difficult to apply the method to the real world where privileged state information is not available. I would be curious to hear the author’s thoughts on this. The limitations section mentions that the same operation could be done on a latent space but its unclear if the RaIR would be applicable to such a latent space (which might not be as good as the privileged information) based state representation.```

---

> ### Author Rebuttal · Authors · 2023-08-09
>
> We thank the reviewer for their feedback and address their questions and comments in the following.
>
> # Clarification on the Graph Architecture
>
> We use a fully-connected Graph Neural Network (GNN).
>
> *How did the authors decide on this specific state representation? Does it have any advantages that a particularly attractive?*
>
> The GNN requires an object-factorized state representation. Note that in this case we take the existing observation in the Construction environment (which is an extension of the original Fetch Pick & Place environment, and has been introduced in [1]). In the original observation, the robot observations (end effector position and velocity, gripper state) and the individual object observations (position, orientation as Euler angles, linear and angular velocities) are concatenated to form the environment observation. We don’t modify the existing state representation from the environment and only factorize this observation vector into its components, which is the format needed for the GNN. The RaIR computation is decoupled from the GNN or any model: it can be done on actual state observation or model predictions. For RaIR, we also use an object-factorized representation, but we don’t feed in the whole state and instead a subspace with x-y-(z) positions.
>
> # Transformers
>
> Indeed, GNNs with attention mechanisms for neighborhood aggregation are shown to be equivalent to transformers with a few modifications. We are planning to test with Transformers as world model backbones in future work as well. It is important to highlight that we expect intrinsic rewards for regularity via RaIR to lead to more capable world models in assembly tasks or in general tasks favoring regularity, irrespective of the world model backbone that is used.
>
> # Policy learning with RaIR
> As RaIR is stationary/Markovian, it can be used as a reward signal to learn a policy via RL.
> Note that for this case, we wouldn’t be training a new policy for each configuration, but we would simply train one policy with RaIR as reward. When we roll out the learnt policy, with the stochasticity of the policy and the different start states in the environment, we would expect different configurations to emerge (i.e. converge to different local minima of RaIR). As such, we don’t see a reason why RaIR wouldn’t scale with RL.
>
> *In Sec. 3.4, why does the agent not arrange everything in a single line to maximize regularity?*
>
> Yes, indeed due to finite-horizon planning as well as the limited sampling budget, we don’t necessarily converge to global minima, which is a full tower for x-y compression and a line (horizontal line or tower) for x-y-z. The planner can find solutions that are reachable within the planning horizon from  the current configuration. For certain starting configurations only local optima are found during an episode.
>
> # Real-world application
> *This state representation might make it difficult to apply the method to the real world where privileged state information is not available.*
>
> We refer the reviewer to our general response, where we discuss this at length.
>
> # Details on Experiment Pipeline
> *How many training steps does it take to train the transition model? This would throw some light on the sample-efficiency of the method. What is “relative time” in Figure 5?*
>
> This is indeed a strong point of our method, which we will explain better in the paper. We refer the reviewer to the general response, where we explain the experiment pipeline and the metrics in detail for further clarification. Here we want to highlight that during the whole free play time, only 600K environment steps are performed (corresponding to about 6.5h of real-world interaction), which underpins the high sample-efficiency.
>
> [1] Li et al (2020), Towards Practical Multi-object Manipulation using Relational Reinforcement Learning. ICRA.

---

> > ### Comment · Reviewer_GveK · 2023-08-13
> > **Thank you for the rebuttal**
> >
> > I thank the authors for the rebuttal and the additional experiments. My concerns have been addressed and it would be great if the authors can include the clarifications in the next version of the paper. I am increasing my score by a point.

---

### Official Review · Reviewer_CLep · 2023-07-08

**Soundness:** 2 fair
**Presentation:** 2 fair
**Contribution:** 2 fair
**Rating:** 6
**Confidence:** 3

**Summary:**

Inspired by the fact that humans prefer symmetric patters, the author proposed to use regularity as intrinsic reward to encourage an agent to discover interesting environmental patterns. One example of regularity is to put boxes in a symmetric pattern. The results showed that this kind of intrinsic reward allows an RL agent to generate more interesting patterns in the absence of task rewards and also allows for better world model learning for zero-shot generalization.

**Strengths:**

- The proposed method is novel and intuitive.
- The experiments are comprehensive, including both analysis of the emergent behaviors and the performance of applying to downstream tasks.

**Weaknesses:**

- Lack of baselines: For section 3.3, I believe RND and ICM can be a reasonable baselines to strengthen the significance of the proposed method.  It seems to me that the goal of section 3.3 is to showcase the practical benefit of RaIR in solving tasks (maximizing extrinsic rewards). To show practical benefit, it would be necessary to answer why the proposed method is a better choice than prior works. Comparison with RND and ICM would answer this question.
- Regularity considered in this paper are not well motivated. I agree the motivation of using regularity as intrinsic rewards is clear, but the specific design of each type of regularity in Table 1 are unclear to me. For example, why we consider these symmetry operations and why design distance in these ways.

**Questions:**

(See weakness)

**Limitations:**

Yes, the limitation is mentioned.

---

> ### Author Rebuttal · Authors · 2023-08-09
>
> We thank the reviewer for their review and feedback. Our response follows:
>
> # Baselines:
>
> In the newly uploaded pdf, we have included experiments with RND. Here we run RND with the same model-based planning backbone (GNN ensemble together with iCEM as in CEE-US) and only alter the intrinsic reward used for planning. (Note that with a version of RND that trains an exploration policy instead of using a planner, meaningful exploration within the budget of 600K interactions is not observed as also reported in [1]).
>
> ICM, on the other hand, computes the intrinsic reward only retrospectively: you need to have visited and observed the actual next state to be able to compute the ICM reward (the error between the model’s prediction and the actual observation). This means that ICM cannot be used in look-ahead planning methods. That’s why we have instead implemented a one-step disagreement version, which is essentially Disagreement from Pathak et.al, 2019 [2]. Here we also used a GNN ensemble backbone with iCEM. (the ICM+exploration policy combination has been shown to not work within the budget of 600K interactions in [1])
>
> Since the experiments are still running, we have included results from the first 180 iterations for our RND variant and 200 iterations of free play for Disagreement. We will update the results once the experiments finish.
> We show that so far both baselines fail to solve the challenging stacking 3 objects task (we plot 3 seeds), in contrast to RaIR+CEE-US and CEE-US with the same amount of free play. Only one seed for RND solves the stacking task with 3% success rate at iteration 180.
>
> Note that we propose RaIR as an additional reward signal to drive exploration towards regularities. As such, RaIR in principle can be used to augment different intrinsic reward functions. In our paper, we illustrated this on the example of CEE-US+RaIR as CEE-US was shown to beat the other baselines (both policy and planning based) in [1].
>
> # Motivation for regularity
> We compute regularity as entropy of the histogram of occurrences for symbols in a multiset. The main question is: how do we obtain these symbols? The general problem is that there is no “ground truth” regularity. In our work, we intended to open the box of regularity that is otherwise not comprehensible: why certain representations lead to certain patterns. Symmetries, as well-characterized regularities, are one way to assess the properties of these representations. This is why we discussed the choice of different mapping functions $\phi$ in terms of their invariances under known symmetry operations. This was intended to be used as an intuitive guide to choose the different representations. For example: since relational RaIR, as opposed to absolute relational RaIR, does not prefer reflections, mostly patterns with translational symmetries are generated, e.g. lines in ShapeGridWorld.
>
> Knowing the properties of these functions and its implications allows us to make design choices to inject certain biases. Similarly, for the recreation of existing patterns we cannot use direct RaIR since it is not invariant to translations. Although perhaps trivial at first glance, we wanted to formalize these properties to provide deeper insight into our regularity reward and its control knobs.
>
> [1] Sancaktar, et al (2022). Curious exploration via structured world models yields zero-shot object manipulation. NeurIPS
>
> [2] Pathak, et al (2019). Self-supervised Exploration via Disagreement, ICML

---

> > ### Comment · Reviewer_CLep · 2023-08-18
> >
> > Thanks for the author's response. The author addresses my questions. I'm increasing my rating to 6.

---

### Author Rebuttal · Authors · 2023-08-09

We thank all reviewers for their constructive feedback and appreciate that they found our method to be "novel and intuitive”, the experiments “comprehensive” with “impressive results for stacking towers in various configurations solely from intrinsic rewards” and the paper “generally well written”.

# Summary
The rebuttal addresses:

**1.** Generality: We added applications of RaIR to new environments (see pdf)
- 2 locomotion environments: Quadruped and Walker
- RoboDesk that contains entities with diverse geometries (drawer, button, blocks etc.)

and modifications of the existing environments:

- Custom Construction with diverse shapes (flat block, short column and ball), where the shapes also differ in their masses.
- ShapeGridWorld with color, where the objects’ colors are included as part of the object feature for regularity.

We showcase the generality of RaIR and its applicability to diverse environments (here using ground truth GT models).

**2.**  Baselines: As suggested, we added two baselines, RND and a type of ICM/ 1-step prediction error, that get outperformed by our method, see Fig D5

**3**  Addressed questions of novelty and object-centric representations

# Baselines
We run two new intrinsic motivation baselines: RND [1] and one-step Disagreement [2] (as a look-ahead alternative to ICM [3] suggested by Reviewer CLep). We include the results on the *Singletower 3* and *Pick & Place 6* tasks. The experiments are still running, thus we include partial results. The baselines solve the pick & place task but completely fail at the more challenging stacking task. Additionally: attempting the 3 stack, RND manages to get to a stack of 2 objects only with $0.39 \pm 0.24$ success rate; one-step disagreement only gets a stack of 2 with  $0.19 \pm 0.02$ success.

# Clarification on object-centric representations

Let us clarify the concerns regarding object-centric representations and interpretable state spaces that were raised by Reviewers fSr3, fHtQ, GveK:

We embrace object-centric representations as a suitable inductive bias in RL, where the observations per object (consisting of poses and velocities) are naturally disentangled. So we don’t see this to be limited. The wording in the paper was perhaps indeed misleading: access to object-centric proprioceptive information is not necessarily privileged or unattainable itself. It is indeed very common in robotics to do pose estimation on objects for control. Approaches such as [5] do object pose estimations from images, and YOLO or even unsupervised object-discovery methods such as Slot Attention have been used to infer object pose, or at least object positions [4]. The assumption that this object-factorized state representation is interpretable is not far-fetched.

The computation of RaIR itself is very generalized. The main design choice goes into which symbols we use to describe the scene and to construct the multiset. In our work, we take a shortcut of putting the human bias in as “compressing positions is a good idea”. But this can be extended: see for instance our new experiments where we put in color as an additional component for regularity (Fig. D1).

Applying RaIR directly to latent representations that are not inherently disentangled presents a challenge: developing a representation mirroring human-relevant structure and regularities. Here, examples of significant regular situations of interest could come in to learn a tokenizable representation for RaIR. This resembles real-world learning, where exposure to regular structures (e.g., towers, bridges) leads us to replicate these patterns while interacting with blocks. However, as Reviewer GveK noted, RaIR computation here faces practical complexities. We will incorporate this discussion per the suggestion in our revised paper.

# Details on Experiment Pipeline

We agree with reviewers GveK and fSr3 that details regarding the terminology and metrics used need better explanation. We provide these here and in the revised paper:
During free play, we start with randomly initialized models and an empty replay buffer. Each iteration of free play (referred to as training iteration in the downstream task success plots) consists of data collection with environment interactions (via online planning), and then model training on the collected data so far (offline).

In each iteration of free play, we collect 2000 samples (20 rollouts with 100 timesteps each) and add them to the replay buffer. During the online planning part for data collection, we only perform inference with the models and no training is performed. Afterwards, we train the model for 25 epochs on the replay buffer. We then continue with data collection in the next free play iteration (see Alg. S1 in appendix).

The relative time is the percentage of timesteps in the collected 2000 samples per free play iteration the agent performs certain types of interactions. So 50% metric for *one object moved* means an object was moved in 1000 timesteps in that free play iteration. *Two or more objects moved* checks if at least 2 objects are moving at the same time. *Object(s) in air* means one or more objects are in air (including being held in air by the agent or being on top of another block). *Object(s) flipped* checks for angular velocities above a threshold for one or more objects, i.e. they are rolled/flipped.
Overall, during the whole free play time, only 600K environment steps are performed, which marks the high sample-efficiency of our method.

[1] Burda et al (2019). Exploration by Random Network Distillation, ICLR

[2] Pathak et al (2019). Self-supervised Exploration via Disagreement, ICML

[3] Pathak et al (2017). Curiosity-driven Exploration by Self-supervised Prediction, ICML

[4] Heravi et al (2023). Visuomotor control in multi-object scenes using object-aware representations, ICRA

[5] Zhang et al (2023). Self-Supervised Geometric Correspondence for Category-Level 6D Object Pose Estimation in the Wild, ICLR

---

> ### Author Response · Authors · 2023-08-21
> **Summary for Part 2 of the Rebuttal Period**
>
> We thank all the reviewers for their responses and their active engagement in the rebuttal.
>
> We want to summarize the experiments we ran and changes we made:
>
> ## Experiments in New Environments
>
> In our first rebuttal update, we showcased the generality of RaIR in new environments by generated patterns and constellations with Ground Truth models.
>
> In the second round of rebuttal, we provided **free play experiments** in the **Generalized Construction** (diverse shapes) and the **Quadruped** environments. We show improved downstream task performance on assembly tasks in the challenging Generalized Construction environment and on Roboyoga poses (tasks from the LEXA Benchmark) in the Quadruped environment. For more details and tables, we refer to our discussion with Reviewer fHtQ. We have also updated our anonymized project webpage https://sites.google.com/view/rair-project (the same link we already had in the main paper submission) with analysis from the free play experiments in the new environments.
>
> ## Baselines
>
> We have now the full results for the two new intrinsic motivation baselines, RND and Disagreement:
>
> | Task@300                   | Singletower 3 | Multitower 2+2 | Pyramid 5   | Pick & Place 6 | Throw 4     | Flip 4      |
> |----------------------------|---------------|----------------|-------------|----------------|-------------|-------------|
> | RaIR+CEE-US                | 0.75 ± 0.07   | 0.77 ± 0.06    | 0.49 ± 0.06 | 0.90 ± 0.02    | 0.32 ± 0.02 | 0.63 ± 0.08 |
> | RaIR                       | 0.64 ± 0.03   | 0.62 ± 0.03    | 0.25 ± 0.05 | 0.74 ± 0.05    | 0.21 ± 0.01 | 0.65 ± 0.1  |
> | CEE-US                     | 0.40 ± 0.12   | 0.52 ± 0.05    | 0.14 ± 0.09 | 0.90 ± 0.02    | 0.49 ± 0.05 | 0.73 ± 0.1  |
> | RND                        | 0.06 ± 0.06   | 0.13 ± 0.11    | 0.01 ± 0.01 | 0.92 ± 0.01    | 0.42 ± 0.08 | 0.74 ± 0.08 |
> | Disagreement (horizon = 1) | 0 ± 0         | 0.01 ± 0       | 0 ± 0       | 0.90 ± 0.0     | 0.31 ± 0.04 | 0.68 ± 0.11 |
>
> **On assembly tasks, RND shows very poor performance and Disagreement fails completely**. For non-assembly tasks, we observe comparable performance for *Pick & Place 6 objects*. And as expected and discussed in the paper, for the more chaos-oriented tasks of *Throwing* and *Flipping*, we see improved performance with CEE-US and RND (Disagreement performance is limited by its planning horizon being 1 timestep during freeplay).
>
> ## Related Work
>
> We have revised our related work section to highlight the distinction of our work from niche-seeking proposed in SMiRL and active inference literature. We also want to highlight that as per Reviewer fHtQ’s request, we have tested SMiRL to showcase that it is indeed **not comparable to RaIR**.
>
> We also discuss the differences between our free play approach with model-based planning vs. goal-conditioned reinforcement learning paradigms.
>
> ## Object-centric Representations
>
> We have also revised the paper to better explain why we think the representation of the world as a collection of entities and their interactions is not very limiting and a productive way forward. However, as we stated in our discussion with Reviewer fHtQ, choosing the right level of abstraction for *entities* in a real-world scenario with composite structures can indeed be challenging with current visual perception pipelines. We have added this to our limitations section.

---

### Decision · Program_Chairs · 2023-09-21

**Decision:**

Accept (poster)

**Comment:**

Exploration is an important problem and the authors suggest a somewhat interesting and novel take on it -- instead of incentivizing novelty, incentivize the agent to form regular structures. All reviewers recommend the paper be accepted, and I concur. Reviewers had concerns about the scope of where the method can apply and I commend the authors for doing a good job of addressing the questions in the rebuttal period.

However, measuring regularity needs strong assumption on the observation space which restricts the applicability of the method. In the rebuttal phase there was ample discussion between the reviewers and the authors and its well summarized here: https://openreview.net/forum?id=BHHrX3CRE1&noteId=033rWTzPdV -- I recommend authors to include additional results and the suggestions here alongwith other suggestions made by the reviewers in the final version of the paper.